# WHICH TASKS SHOULD BE COMPRESSED TOGETHER? A CAUSAL DISCOVERY APPROACH FOR EFFICIENT MULTI-TASK REPRESENTATION COMPRESSION

**Sha Guo,**[1]     **Jing Chen,**[1]     **Zixuan Hu,**[1]     **Zhuo Chen,**[2]     **Wenhan Yang,**[2]
**Yu Lin,**[3]     **Xing Jiang,**[3]     **Ling-Yu Duan**[1,2*]
[1]School of Computer Science, Peking University   [2]Peng Cheng Laboratory, Shenzhen, China
[3] Fuzhou Chengtou New Infrastructure Group, Fuzhou, China
`sandykwokcs@stu.pku.edu.cn, chenjing15@pku.org.cn, hzxuan@pku.edu.cn,`
`{chenzh08, yangwh}@pcl.ac.cn, {linyu2679, jiangxing315}@gmail.com,`
`lingyu@pku.edu.cn`

## ABSTRACT

Conventional image compression methods are inadequate for intelligent analysis, as they overemphasize pixel-level precision while neglecting semantic significance and the interaction among multiple tasks. This paper introduces a Taskonomy-Aware Multi-Task Compression framework comprising (1) inter-coherent task grouping, which organizes synergistic tasks into shared representations to improve multi-task accuracy and reduce encoding volume, and (2) a conditional entropy-based directed acyclic graph (DAG) that captures causal dependencies among grouped representations. By leveraging parent representations as contextual priors for child representations, the framework effectively utilizes cross-task information to improve entropy model accuracy. Experiments on diverse vision tasks, including Keypoint 2D, Depth Z-buffer, Semantic Segmentation, Surface Normal, Edge Texture, and Autoencoder, demonstrate significant bitrate-performance gains, validating the method's capability to reduce system entropy uncertainty. These findings underscore the potential of leveraging representation disentanglement, synergy, and causal modeling to learn compact representations, which enable efficient multi-task compression in intelligent systems.

## 1 INTRODUCTION

Multimodal models like CLIP (Radford et al., 2021), GPT-4 (Achiam et al., 2023), and Sora (Liu et al., 2024a) exhibit human-level comprehension and reasoning (Achiam et al., 2023; Chang et al., 2024; Zheng et al., 2023; Laskar et al., 2023), making them potential consumers of visual and multimedia content. This highlights the need for semantic representation compression to support efficient multi-task processing. However, current compression techniques, including both handcrafted video codecs (Bross et al., 2021; Pennebaker & Mitchell, 1992; Si & Shen, 2016) and end-to-end learning-based approaches (Jiang et al., 2023; He et al., 2022; Zou et al., 2022; Chen et al., 2021; Ballé et al., 2017), primarily focus on rate-distortion optimization (Shannon et al., 1959). Handcrafted methods typically rely on traditional techniques to minimize redundancy and maintain visual quality, whereas end-to-end learning-based methods often involve constraining the entropy model to accurately estimate the probability distribution of latent space symbols while simultaneously maximizing the pixel-level likelihood between the reconstructed and original images. Nevertheless, conventional compression methods lack semantic representation constraints, limiting their ability to preserve task-relevant information while reducing redundancy. As illustrated in Fig. 1a, assessment on MS-COCO (Lin et al., 2014) tasks with images compressed at 0.15 bpp shows learning-based methods MLIC++ (Jiang et al., 2023) and ELIC (He et al., 2022) exhibit a notable superiority compared to the handcrafted WebP (Si & Shen, 2016) and VTM-17.0 (Bross et al., 2021), particularly in multimodal tasks like Video Question Answering (Zhang et al., 2023). This advantage can be further enhanced by task-specific auxiliary loss functions, highlighting the necessity to capture richer semantics for intelligent multimedia analysis.

---

*Corresponding Author.

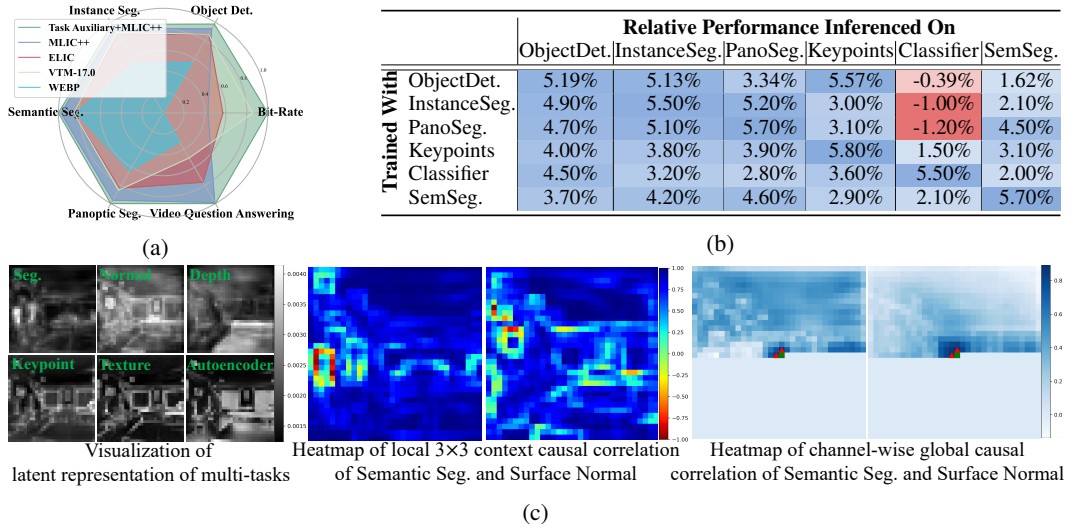

Figure 1: **Motivation of Taskonomy-Aware Multi-Task Compression. (a)** *Multi-task Compression Performance*: Normalized performance (0=worst, 1=best for clarity) of compressors in diverse tasks. **(b)** *Necessity of Task Grouping*: Relative performance (%) when compressors are trained on one task and tested on another. Positive values indicate collaboration, and negative values highlight conflicts, underscoring the need for task grouping. **(c)** *Necessity of Taskonomy-Aware Causal Modeling*: Semantic Seg. and Surface Normal tasks' local context correlation (middle) and global channel-wise correlation (right) are visualized, showing the potential for capturing redundancy and achieving bit savings through task-aware causal modeling. The ■ indicates the current decoding point, while the ▲ indicates the most similar reference positions.

In response to these limitations, the paradigm of Video Coding for Machines (VCM) (Yang et al., 2024; Choi & Bajić, 2022; Duan et al., 2020) has emerged as a promising solution. VCM integrates image compression with feature representation to achieve both compactness and efficiency, aiming to meet the dual objectives of high-fidelity human vision and high-precision machine vision (Ge et al., 2024; Li et al., 2024; Liu et al., 2023; Bai et al., 2022).

Despite advances in VCM, most methods still treat tasks in isolation (Liu et al., 2021; Bai et al., 2022) or focus only on predefined tasks (Liu et al., 2023; Li et al., 2024), overlooking the benefits of grouping supportive tasks or complex relationships between feature spaces of tasks. Preliminary studies (Shi et al., 2023; Fifty et al., 2021; Standley et al., 2020; Zamir et al., 2018) and our findings in Fig. 1b reveal statistically significant correlations among tasks, both positive and negative. These correlations highlight the synergies and conflicts in multi-task compression. Therefore, identifying task groups that leverage synergies and reduce conflicts is essential for improved multi-task learning and optimized compression performance. Furthermore, Fig. 1c demonstrates the existence of local and global causal contextual relationships across tasks with different semantic granularities. Exploiting conditional relationships in representations enables more accurate prediction of symbol distributions through conditional entropy, thereby improving compression efficiency. Conventional methods, however, process tasks independently, ignoring inter-task dependencies, which leads to redundancy.

Thus, an essential question arises: *How can we discern and exploit the interdependencies among tasks to achieve efficient multi-task representation compression?* Addressing this question requires models that can (1) discern mutually beneficial and conflicting tasks to group them effectively, and (2) dynamically decouple and model the dependencies between sub-tasks, enabling independent optimization of each sub-task while preserving the hierarchical task relationships.

To address this challenge, we propose a paradigm shift with *Taskonomy-Aware Multi-Task Compression* (TAMC), which integrates task grouping and causal discovery for compact multi-task representation compression. By leveraging causal relationships between tasks, TAMC enhances task performance and improves overall compression efficiency. Our approach consists of two key components. First, we cluster inter-coherent tasks into groups that share a universal representation, leveraging synergies among mutually beneficial tasks to improve accuracy and reduce encoding

volume. This is the first work to systematically group tasks for compression, mitigating conflicts and preventing performance degradation. Second, we construct a directed acyclic graph (DAG) based on conditional entropy to capture causal relationships among task representations. This approach identifies task dependencies across different abstraction levels, uncovering inter-task relationships and mapping information flow through directed graphs. By traversing causal paths, parent task representations provide informative cross-task contexts for child tasks, thus reducing uncertainty and enhancing compression efficiency and scalability.

Our contributions: 1. We demonstrate that leveraging intricate inter-task relationships significantly improves rate-performance efficiency by clustering tasks for collective compression and establishing causal links between clusters. 2. We introduce a DAG-based causal discovery framework via conditional entropy, which captures semantic dependencies across abstraction levels to enhance system certainty and reduce information entropy, thereby improving compression compactness. 3. The proposed TAMC achieves superior performance across diverse downstream task benchmarks while remaining competitive in universal image reconstruction. Extensive experiments on key computer vision tasks using the Taskonomy dataset validate the effectiveness of our approach.

## 2 RELATED WORK

**Multi-Task Learning.** Multi-task learning (MTL) improves performance by introducing inductive biases and emphasizing relevant features (Zhang & Yang, 2021). However, task competition for model capacity and ineffective shared representations often hinder MTL. Grouping compatible tasks is crucial for reducing conflicts and boosting performance (Lu et al., 2020; Yu et al., 2020; Chen et al., 2020; Kendall et al., 2018), yet current approaches often rely on human intuition (Zhang & Yang, 2021). Recent studies (Fifty et al., 2021; Wu et al., 2020; Standley et al., 2020) highlight the need for systematic task grouping to advance the field.

**End-to-End Image Compression for Human and Machine Tasks.** In E2E-learned image compression (Ballé et al., 2017), an image $x$ is first encoded into latent representations $y$ using an analysis transform $g_a(x; \theta_e)$, then quantized to discrete values $\hat{y}$. With a learned probability model $p_{\hat{y}}(\hat{y})$, $\hat{y}$ can be losslessly coded using arithmetic coding. On the decoder side, a synthesis transform $g_s(\hat{y}; \theta_d)$ reconstructs the image $\hat{x}$ from $\hat{y}$:

$$y = g_a(x; \theta_e), \hat{y} = Q(y), \hat{x} = g_s(\hat{y}; \theta_d). \tag{1}$$

To improve compression efficiency by decorrelating the latent space and estimating symbol probabilities, Ballé et al. (2018) introduces a hyperprior model that reduces spatial redundancies among latent variables, adding a few extra bits to convey spatial structure. This hyperprior model enables a more accurate entropy model and better estimation of $p_{\hat{y}}(\hat{y})$. It can be divided into a hyper analysis transform $h_a(y; \theta_{he})$ and a synthesis transform $h_s(\hat{z}; \theta_{hd})$:

$$z = h_a(y; \theta_{he}), \hat{z} = Q(z), p_{\hat{y}|\hat{z}}(\hat{y}|\hat{z}) = h_s(\hat{z}; \theta_{hd}). \tag{2}$$

Minnen et al. (2018) proposed a more accurate entropy model which jointly utilizes an autoregressive context model $g_{cm}$. The predicted Gaussian parameters $N(\mu, \sigma)$ of the distribution $p_{\hat{y}}(\hat{y})$ are functions of the learned parameters of the hyper-decoder, context model, and entropy parameter networks ($\theta_{hd}$, $\theta_{cm}$, and $\theta_{ep}$, respectively):

$$p_{\hat{y}}(\hat{y} \mid \hat{z}, \theta_{hd}, \theta_{cm}, \theta_{ep}) = \prod_i \left( \mathcal{N}(\mu_i, \sigma_i^2) * \mathcal{U}\left(-\frac{1}{2}, \frac{1}{2}\right) \right)(\hat{y}_i), \tag{3}$$

with $\mu_i, \sigma_i = g_{ep}(\psi, \phi_i; \theta_{ep})$, $\psi = g_h(\hat{z}; \theta_{hd})$, and $\phi_i = g_{cm}(\hat{y}_{<i}; \theta_{cm})$, $\mathcal{U}\left(-\frac{1}{2}, \frac{1}{2}\right)$ is a uniform noise to approximate quantization during training. The overall loss function is:

$$\mathcal{L} = \mathcal{R}(\hat{y}) + \mathcal{R}(\hat{z}) + \lambda \cdot \mathcal{D}(x, \hat{x}) = \underbrace{\mathbb{E}\left[-\log_2(p_{\hat{y}|\hat{z}}(\hat{y}|\hat{z}))\right]}_{\text{rate(latents)}} + \underbrace{\mathbb{E}\left[-\log_2(p_{\hat{z}}(\hat{z}))\right]}_{\text{rate(hyper-latents)}} + \lambda \cdot \underbrace{\mathcal{D}(x, \hat{x})}_{\text{distortion}}, \tag{4}$$

where $\lambda$ controls the rate-distortion tradeoff. The first term is the rate that corresponds to the cross entropy between the natural (marginal) distribution and the learned entropy model. The second term is the rate to transmit hyperprior. The third term measures the reconstruction quality according to the given distortion metric $d$ (*e.g.*, PSNR or MS-SSIM). Recent advancements with improved entropy

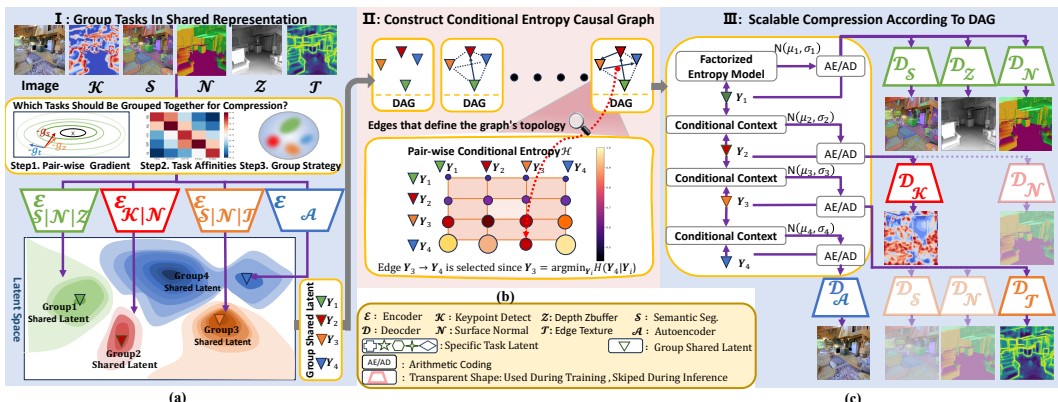

Figure 2: Overview of TAMC. **Given a series of tasks, how can we effectively cluster them in the latent semantic space and construct a causal graph to optimize bitrate-performance?** To find a feasible solution, we follow these steps: (**I**): Group tasks based on inter-task coherence under bitrate constraints. (**II**): Construct a DAG via conditional entropy to capture causal relationships. (**III**): Compress grouped representations according to DAG in a scalable manner.

models (*e.g.*, hierarchical (Ballé et al., 2018), auto-regressive (Minnen et al., 2018; He et al., 2021; Xiang et al., 2022)), multireference entropy (Qian et al., 2020), and innovations such as channel-wise and spatial-wise acceleration (Minnen & Singh, 2020; He et al., 2021; Jiang et al., 2023; He et al., 2022), codebooks and vector quantization (Zhu et al., 2022), and hierarchical VAEs (Duan et al., 2023), have significantly improved compression performance. Many existing techniques overlook their impact on downstream tasks like classification, detection, and segmentation, which require task-specific feature retention. Task-aware paradigms such as Video Coding for Machines (VCM) (Yang et al., 2024; Choi & Bajić, 2022) focus on machine vision-targeted compression(Li et al., 2024; Liu et al., 2023), including *feature-assisted coding* (Liu et al., 2024b), *scalable coding*(Liu et al., 2021), and *intermediate feature compression*(Kim et al., 2023; Chen et al., 2021). While these methods are effective for individual tasks(Liu et al., 2021; Li et al., 2024; Liu et al., 2023), they often fall short in capturing the full complexity of multi-task relationships, highlighting the need for more effective multi-task compression strategies.

## 3 APPROACH

### 3.1 ARCHITECTURE

An overview of TAMC is provided in Fig. 2. It comprises three key components: (1) a *inter-coherent task cluster* that groups mutually coherent tasks into a shared representation space; (2) a *conditional entropy graph*, constructed using causal discovery to reveal dependencies; and (3) a *scalable compressor*, which compresses multiple feature layers along graph paths.

Components (1) and (2) draw inspiration from the lookahead stage in traditional video encoding (Li, 2003; He & Mitra, 2002; Wang & Kwong, 2008; Ma et al., 2005), which performs preliminary analysis to optimize the encoding performance of Component (3) under bitrate constraints. In Component (1), $2\times$ downsampled low-resolution images are used as inputs, combined with a low-complexity feature extraction and decoding backbone (Standley et al., 2020), to efficiently pre-analyze task grouping performance under the constraint of bitrate consumption. The task grouping strategy leverages gradient coherence to cluster tasks, maximizing shared information while minimizing redundancy. In Component (2), a conditional entropy graph organizes the grouped tasks hierarchically, facilitating information transfer from parent tasks to child tasks. This hierarchical structure improves representation certainty and encoding efficiency by providing a more precise cross-task entropy estimation model.

### 3.2 GROUP INTER-COHERENT TASKS

The objective of task grouping is to cluster tasks that can share representations, thereby optimizing the performance of multiple tasks under a given bitrate budget $b$. As task grouping is conducted in

the pre-analysis module, the bitrate budget $b$ is loosely approximated $\propto$ the number of task groups.[1]. Formally, a set of $n$ tasks $\mathcal{T} = \{\tau_1, \ldots, \tau_n\}$, which is partitioned into $k$ subgroups $\mathcal{C} = \{c_1, \ldots, c_k\}$, where each task $\tau_i \in \mathcal{T}$ belongs to at least one subgroup, *i.e.*, $\forall \tau \in \mathcal{T}, \exists c_g \in \mathcal{C} \mid \tau \in c_g$.[2]. $k \leq n$ and each subgroup $c_g \subseteq \mathcal{T}$ shares a feature extractor (encoder) $g_a(\boldsymbol{x}; \boldsymbol{\theta}_e)$, as defined in Eq. 1. To distinguish between the lookahead and formal encoding stages, the set of shared feature extractors is denoted as $\mathcal{E} = \{l_{a_1}, \ldots, l_{a_k}\}$. For a raw image $\boldsymbol{x}$, the encoder $l_{a_g}$ in subgroup $c_g$ generates a shared representation $\boldsymbol{y}_g = l_{a_g}(\boldsymbol{x}; \boldsymbol{\theta}_{e_g})$. Each task $\tau_i$ is associated with a unique decoder $l_{s_i}(\boldsymbol{y}; \boldsymbol{\theta}_{d_i})$, corresponding to the synthesis transform $g_s(\hat{\boldsymbol{y}}; \boldsymbol{\theta}_d)$ in Eq. 1. The decoders $D = \{l_{s_1}, l_{s_2}, \ldots, l_{s_n}\}$ map the latent feature $\boldsymbol{y}_{g(i)}$ into task space, where $g(i)$ denotes the subgroup index for $\tau_i$. The predicted output $\hat{\boldsymbol{x}}_i = l_{s_i}(\boldsymbol{y}_{g(i)}; \boldsymbol{\theta}_{d_i})$ is then compared with the ground truth for evaluation.

Task performance of $\tau_i$ is measured using task-specific loss functions $L_i$, *e.g.*, PSNR/MS-SSIM for pixel reconstruction and cross-entropy loss for segmentation. For a given grouping strategy $\mathcal{C} = \{c_1, \ldots, c_k\}$, the overall performance of the task grouping is aggregated as $\sum_{i=1}^{n} \min_g \left( \mathbb{I}(\tau_i \in c_g) L_i \right)$.The bitrate $\mathcal{R}$ required to transmit the latent feature $\boldsymbol{y}_g$ for each group $c_g$ is approximated by $b_g = B(\boldsymbol{y}_g)$, where $B$ is $\propto$ the amount of data transmitted, with each shared representation consuming a unit cost of 1. Our goal is to minimize the total loss subject to the total bitrate constraint $b$:

$$\mathcal{L} = \sum_{i=1}^{n} \lambda_i \min_g \left( \mathbb{I}(\tau_i \in c_g) L_i \right) + \sum_{j=1}^{k} B(\boldsymbol{y}_j), \quad \text{s.t.} \sum_{j=1}^{k} B(\boldsymbol{y}_j) \leq b. \tag{5}$$

To achieve this, we first analyze the influence of task gradients on each other to determine the subgroup formation. Task grouping is then guided by gradient consistency, ensuring effective collaboration among tasks. By sharing gradient updates, encoding parameters, and feature representations, tasks within the same group enhance mutual learning. The specific methodology is as follows:

Consider a multi-task model that trains all tasks together with parameters $\Theta_g \cup \{\Theta_i | i \in \mathcal{T}\}$, where $\Theta_g = \{\boldsymbol{\theta}_{e_g}\}$ denotes shared parameters for all tasks, and $\Theta_i = \{\boldsymbol{\theta}_{d_i}\}$ represents task-specific parameters for $\tau_i$. Given input images $\mathcal{X}$, we optimize the overall task performance by following the total loss $\mathcal{L}_{total}$:

$$\mathcal{L}_{total} = \sum_{i \in \mathcal{T}} L_i(\mathcal{X}; \Theta_g, \Theta_i). \tag{6}$$

At the time step $t$, for the input batch $\boldsymbol{x}^t$, the gradient descent update for the shared parameters $\Theta_g$ with respect to $\tau_u$ at step $t+1$ is denoted as $\Theta_{g|u}^{t+1}$, and is computed as follows:

$$\Theta_{g|u}^{t+1} \leftarrow \Theta_g^t - \eta \nabla_{\Theta_g^t} L_u(\boldsymbol{x}^t; \Theta_g^t, \Theta_u^t). \tag{7}$$

Using these updated shared parameters $\Theta_{g|u}^{t+1}$, we calculate the forecast loss for other tasks $\tau_v$ while keeping task-specific parameters $\Theta_v^t$ and inputs $\boldsymbol{x}^t$ unchanged. Gradient coherence between tasks $\tau_u$ and $\tau_v$ is then measured as:

$$C_{u \to v}^t = 1 - \frac{L_v(\boldsymbol{x}^t; \Theta_{g|u}^{t+1}, \Theta_v^t)}{L_v(\boldsymbol{x}^t; \Theta_g^t, \Theta_v^t)}. \tag{8}$$

A positive $C_{u \to v}^t$ indicates that the update from $\tau_u$ reduces the loss for $\tau_v$, while a negative value $C_{u \to v}^t$ value indicates conflicting parameter update directions. Tasks with high gradient coherence are grouped to share encoders and representations. After $T$ timesteps, the cumulative coherent measure is calculated as: $\hat{C}_{u \to v} = \frac{1}{T} \sum_{t=1}^{T} C_{u \to v}^t$. After obtaining the gradient coherence measures between all pairs of tasks, the number of possible grouping for $n$ tasks is $2^n - 1$, which grows exponentially with $n$. To address this computational complexity, we apply relaxed estimates for higher-order cluster coherent measures. Specifically, for a triplet of tasks $\{\tau_u, \tau_v, \tau_w\}$, the triplet cost is estimated as the average of pairwise coherence measures $\hat{C}_{u \to v}$ and $\hat{C}_{w \to v}$. Assuming each cluster has a unit bitrate cost, we adopt a branch-and-bound method, as in prior works (Fifty et al., 2021; Standley et al., 2020; Zamir et al., 2018), to search for optimal grouping strategies under the given bitrate budget $b$.

---

[1]Note: In the E2E compression training phase, bitrate consumption is defined as $\mathbb{E}_{\boldsymbol{x} \sim p_{\boldsymbol{x}}} \left[ -\log_2 p_{\hat{\boldsymbol{y}}}(\hat{\boldsymbol{y}}) \right]$. During deployment, the cumulative distribution function (CDF) of $P_{\hat{\boldsymbol{y}}}(\hat{\boldsymbol{y}})$ is used for arithmetic coding to determine the final bitrate.

[2]As shown in Fig. 2, tasks may belong to multiple groups. Transparent tasks are used during training to boost group performance but are discarded during inference.

## 3.3 CAUSAL GRAPH-BASED COMPRESSION METHOD

Building upon the task grouping strategies and the group-shared variables $\mathcal{Y}$, we now focus on how to construct the directed acyclic graph (DAG) that represents the causal relationships among these variables. Let $\mathcal{Y} = \{Y_1, Y_2, \ldots, Y_k\}$ be a set of group-shared random variables, where each $Y_i$ corresponds to the shared representation $\boldsymbol{y}_i$ for task group $c_i$. We say that $Y_i$ *causes* $Y_j$ if there exists a function $f_j$ such that $Y_j = f(Y_i, C_j)$, where $C_j$ represents the contextual information influencing $Y_j$, including both observable and latent factors. This causal relationship is represented by the edge $Y_i \rightarrow Y_j$ in a directed acyclic graph (DAG) $G = (V, E)$, where $V$ denotes the vertex (*i.e.*, $\mathcal{Y}$ in our context) and $E$ denotes the set of edges.

---

**Algorithm 1** : DAG Construction via Conditional Entropy

---

**Require:** Latents set $\mathcal{Y} = \{Y_1, \ldots, Y_k\}$, conditional entropy oracle $\mathscr{H}$
**Ensure:** Returns the causal DAG $G = (V, E)$
 1: Initialize $V = \mathcal{Y}$ and $E = \emptyset$
 2: $\mathcal{Y} = \text{Sort}(\mathcal{Y}, \text{key} = H(Y), \text{reverse} = \text{True})$
 3: **for** each latent $Y_c \in \mathcal{Y}$ **do**
 4:     Initialize $\min H(Y_c|\cdot) \leftarrow \infty, Y_p \leftarrow$ None
 5:     **for** each latent $Y_j \in \mathcal{Y} \setminus \{Y_c\}, j > c$ **do**
 6:         Look up $H(Y_c|Y_j)$ from $\mathscr{H}$
 7:         **if** $H(Y_c|Y_j) < \min H(Y_c|\cdot)$ **then**
 8:             Update $\min H(Y_c|\cdot) \leftarrow H(Y_c|Y_j)$
 9:             Update $Y_p \leftarrow Y_j$
10:         **end if**
11:     **end for**
12:     **if** $Y_p \neq$ None **then**
13:         Add edge $Y_p \rightarrow Y_c$ to $E$
14:     **end if**
15: **end for**
16: **return** $G = (V, E)$

---

Identifying the true causal graph without experiments or strong assumptions is generally infeasible. To efficiently compress the representations $\mathcal{Y}$, we approximate causal relationships by minimizing the conditional entropy, which quantifies the uncertainty of one variable given another. Specifically, given $Y_i$ and $Y_j$ with domains $\mathcal{Y}_i$ and $\mathcal{Y}_j$ respectively, the pair-wise conditional entropy $H(Y_i|Y_j)$ (also denoted as the conditional entropy oracle $\mathscr{H}$ in **Algorithm 1**) is obtained using the Kernel Density Estimator (KDE) from (Choi & Bajić, 2022; Saxe et al., 2019), and defined as:

$$H(Y_i|Y_j) = H(Y_i) - I(Y_i; Y_j) = -\sum_{\boldsymbol{y}_i} p(\boldsymbol{y}_i) \log p(\boldsymbol{y}_i) - \sum_{\boldsymbol{y}_i, \boldsymbol{y}_j} p(\boldsymbol{y}_i, \boldsymbol{y}_j) \log \frac{p(\boldsymbol{y}_i, \boldsymbol{y}_j)}{p(\boldsymbol{y}_i) p(\boldsymbol{y}_j)}. \tag{9}$$

We define the parent $Y_p$ of the children variable $Y_c$ as the one that minimizes the conditional entropy:

$$Y_p = \arg \min_{Y_j \in \mathcal{Y} \setminus \{Y_c\}, H(Y_j) < H(Y_c)} H(Y_c|Y_j). \tag{10}$$

By applying this criterion iteratively to all representations in $\mathcal{Y}$, we construct the edges $E$ of the causal graph $G$. The causal discovery process is detailed in **Algorithm** 1.

**Scalable Compression Using the Causal DAG**. After constructing the directed acyclic graph (DAG) $G$, inspired by the causal context entropy model Guo et al. (2021) and MLIC++ (Jiang et al., 2023), we perform compression by traversing the graph in topological order. The parent representation $\boldsymbol{y}_p$ serves as an additional cross-task context for the child representation $\boldsymbol{y}_c$, thereby improving compression efficiency. Specifically, when the parent latent representation $\boldsymbol{y}_p$ is assumed to *cause* the child representation $\boldsymbol{y}_c$, the distribution of $\boldsymbol{y}_p$ is modeled using the prior framework in Eq. 3. The estimation of $p_{\hat{\boldsymbol{y}}_p}(\hat{\boldsymbol{y}}_p)$ is performed using a masked convolution $\phi_{p,i} = g_{cm}(\hat{\boldsymbol{y}}_p < i; \boldsymbol{\theta}_{cm})$, which extracts the local context $\phi_{p,i}$. This context is then integrated with the hyperpriors $\boldsymbol{z}_p$ to estimate the Gaussian distribution parameters for $\hat{\boldsymbol{y}}_p$. When it comes to ecoding $\hat{\boldsymbol{y}}_c$, we aggregate both the decoded latent $\hat{\boldsymbol{y}}_p$ and the first half latent in the current spatial location $\hat{\boldsymbol{y}}_c$, generating more informative contexts $\phi_{c,i}$. As shown in Fig. 3a, the whole process can be extended from Eq.3 and formulated as:

$$p_{\hat{\boldsymbol{y}}_c}(\hat{\boldsymbol{y}}_c \mid \hat{\boldsymbol{z}}_c, \boldsymbol{\theta}_{hd}, \boldsymbol{\theta}_{cm+}, \boldsymbol{\theta}_{ep}) = \prod_i \left( \mathcal{N}(\mu_{c,i}, \sigma_{c,i}^2) * \mathcal{U}\left(-\frac{1}{2}, \frac{1}{2}\right) \right) (\hat{\boldsymbol{y}}_{c,i}), \tag{11}$$

with $\mu_{c,i}, \sigma_{c,i} = g_{ep}(\boldsymbol{\psi}_c, \phi_{c,i}; \boldsymbol{\theta}_{ep}), \boldsymbol{\psi}_c = g_h(\hat{\boldsymbol{z}}_c; \boldsymbol{\theta}_{hd})$ , and $\phi_{c,i} = g_{cm+}(\hat{\boldsymbol{y}}_c < i, \hat{\boldsymbol{y}}_p; \boldsymbol{\theta}_{cm+})$. The differences from Eq. 3 are highlighted in blue.

The above causal context model primarily captures local correlations while ignoring long-range dependencies. To address this limitation, we utilized a *causal global prediction model*, which leverages long-range correlations between the parent $\hat{\boldsymbol{y}}_p$ and child $\hat{\boldsymbol{y}}_c$. The overall process shown in

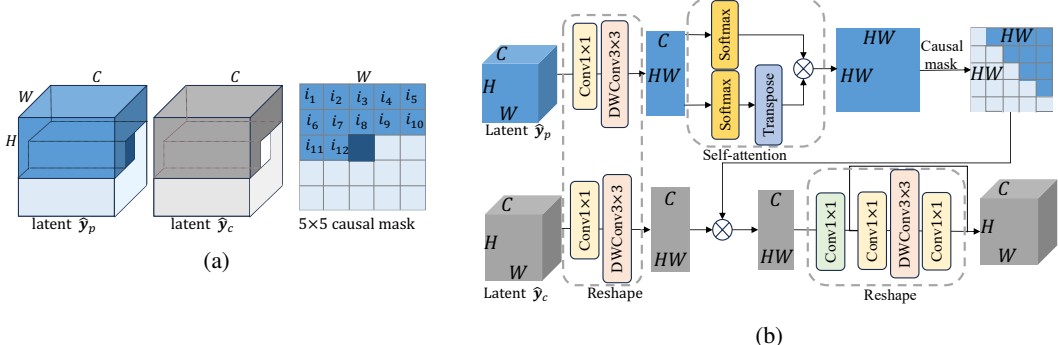

Figure 3: The structure of proposed cross-task causal context compression: (a) Cross-task local causal context mask convolution. (b) The parent $\hat{\boldsymbol{y}}_p$ provides a global causal context for the child $\hat{\boldsymbol{y}}_c$.

Fig.3b and is formulated as:

$$\hat{\boldsymbol{y}}_{c,i}^{\text{attn}} = \underbrace{\text{softmax}_2\left(\hat{\boldsymbol{y}}_p^q < i\right)\text{softmax}_1\left(\hat{\boldsymbol{y}}_p^k < i\right)^\top \hat{\boldsymbol{y}}_c^v < i}_{\text{non-negative}}, \hat{\boldsymbol{y}}_{c,i}^{\text{conv}} = \text{conv}_{K\times K}(\hat{\boldsymbol{y}}_{c,i}^{\text{attn}}), \boldsymbol{\phi}_{gc,i} = \text{DepthRB}(\hat{\boldsymbol{y}}_{c,i}^{\text{conv}}),$$

(12)

where $\hat{\boldsymbol{y}}_p^q < i$, $\hat{\boldsymbol{y}}_p^k < i = \text{Embedding}(\hat{\boldsymbol{y}}_p < i)$, $\hat{\boldsymbol{y}}_c^v < i = \text{Embedding}(\hat{\boldsymbol{y}}_c < i)$. The Embedding layer consists of a $1 \times 1$ convolutional layer and a $3 \times 3$ depth-wise convolutional layer. The $3 \times 3$ depth-wise convolutional layer is employed for learnable position embedding. DepthRB is proposed by (Jiang et al., 2023) to enhance the non-linearity. Denoting the trainable parameters in the causal global prediction model as $\boldsymbol{\theta}_{gc}$, Eq.3 is extended as:

$$p_{\hat{\boldsymbol{y}}_c}(\hat{\boldsymbol{y}}_c \mid \hat{\boldsymbol{z}}_c, \boldsymbol{\theta}_{hd}, \boldsymbol{\theta}_{cm}, \boldsymbol{\theta}_{gc}, \boldsymbol{\theta}_{ep}) = \prod_i \left( \mathcal{N}(\mu_{c,i}, \sigma_{c,i}^2) * \mathcal{U}\left(-\frac{1}{2}, \frac{1}{2}\right)\right)(\hat{\boldsymbol{y}}_{c,i}),$$

(13)

with $\mu_{c,i}, \sigma_{c,i} = g_{ep}(\boldsymbol{\psi}_c, \boldsymbol{\phi}_{c,i}; \boldsymbol{\phi}_{gc,i}; \boldsymbol{\theta}_{ep})$, $\boldsymbol{\psi}_c = g_h(\hat{\boldsymbol{z}}_c; \boldsymbol{\theta}_{hd})$, and $\boldsymbol{\phi}_{c,i} = g_{cm}(\hat{\boldsymbol{y}}_c < i, \hat{\boldsymbol{y}}_p; \boldsymbol{\theta}_{cm})$, $\boldsymbol{\phi}_{gc,i} = g_{gc}(\hat{\boldsymbol{y}}_c < i, \hat{\boldsymbol{y}}_p; \boldsymbol{\theta}_{gc})$. The differences from Eq. 11 are highlighted in green.

## 4 EXPERIMENTS

### 4.1 TASKS AND DATASETS

To quantify the performance across diverse downstream tasks, we evaluate on the Taskonomy dataset (Zamir et al., 2018) across 6 tasks. Taskonomy is a large-scale computer vision dataset that includes over 4.5 million images from more than 500 buildings. Each image has 18 annotations covering 2D, 3D, and semantic tasks. The total size of the dataset is 11.16 TB. Due to limited computational and storage resources, we used the Tiny split for our experiments, which consists of 872,517 images in the training set and 16,000 images in the validation set and test set, respectively. We conducted experiments on 6 tasks selected from the 15 annotated tasks, *i.e.*, Semantic Segment, Keypoint 2D, Edge Texture, Surface Normal, Depth Z-buffer and Autoencoder. More details of tasks and loss measurements are provided in A.2.

### 4.2 BASELINES

Our method is compared against several baselines, including traditional codecs such as JPEG (Pennebaker & Mitchell, 1992), WebP (Si & Shen, 2016), and VTM-17.0 (Bross et al., 2021), as well as learning-based compression methods, *i.e.*, ELIC (He et al., 2022) and MLIC++ (Jiang et al., 2023).

To evaluate the performance of different compression methods across a variety of tasks, we utilized the official open-source implementations of these methods to compress input images. The reconstructed images were then evaluated on multiple downstream tasks by analyzing the features extracted by pre-trained task models provided by (Standley et al., 2020) [3].

---

[3] https://drive.google.com/drive/folders/1XQVpv6Yyz5CRGNxetO0LTXuTvMS_w5R5?usp=sharing

For our method, we first employed Xception (Chollet, 2017) as the encoder backbone, and the task-specific decoder consists of four transposed convolutional layers and four convolutional layers. Initially, we trained the group-task shared encoder-decoder without compression loss for 60 epochs using the SGD optimizer (Ruder, 2016), with the learning rate decaying from 0.1 to 1e-4. Then, based on the conditional entropy of the shared representation, we constructed a directed acyclic graph (DAG) among the child nodes of the shared representation. Finally, we learned entropy models for different compression rates by following paths from parent nodes to child nodes. We set the task learning rate to 1e-4, and the learning rate of the hyperprior entropy model to 1e-4. We continued training for 50 epochs, adjusting the $\lambda$ parameter in the distortion-rate trade-off $\lambda \times \mathcal{D} + \mathcal{R}$ with values from [0.04, 0.072, 0.14, 1, 1.932] to learn encoder-decoder models parameters and entropy model parameters for different bitrates.

## 4.3 RESULTS

### 4.3.1 PERFORMANCE OF COMPRESSION FOR MULTIPLE TASKS

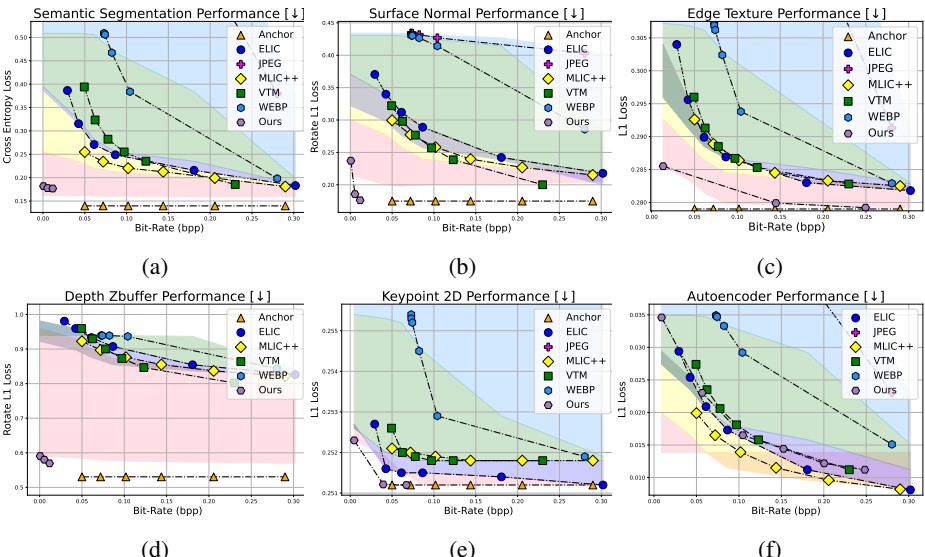

Figure 4: Comparison of performance-rate curves for 6 tasksusing baseline compression methods and our proposed TAMC. "Anchor" refers to the optimal performance of a supervision task obtained using uncompressed images as input. The area of the shade color visualizes bitrate-performance gains.

Our results, summarized in Fig. 4, demonstrate that TAMC achieves a new state-of-the-art in compression for multitask scenarios, significantly outperforming other baselines in 5 out of 6 tasks. In multitask learning and compression, different tasks impose distinct requirements on image information. For example, Depth Z-buffer in Fig. 4d relies on geometric details. Therefore, effective compression for Depth Z-buffer must retain global structural cues to preserve depth continuity. In contrast, Surface Normal estimation in Fig. 4b and Edge Texture detection in Fig. 4c primarily depend on local sharp edges and fine-grained texture. Consequently, compression strategies for these tasks should prioritize preserving high-frequency details rather than global structure. Traditional compression methods (*i.e.*, JPEG, WebP, VTM) and end-to-end deep learning approaches (*i.e.*, ELIC, MLIC++) typically use *a unified compression representation for all tasks, overlooking the uniqueness, conflicts, and collaboration among multiple tasks*. This often results in suboptimal performance for certain tasks. A more detailed analysis is provided in A.3.1. TAMC addresses this limitation by first partitioning tasks into complementary groups, ensuring that collaborative tasks within the same group share representations. We then employ causal discovery through conditional entropy to identify dependencies among groups. These shared representations are scalably compressed following parent-child relationships, effectively leveraging the context priors from parent nodes to reduce the uncertainty of child nodes. A more detailed analysis is provided in A.3.2.

### 4.3.2 PERFORMANCE OF IMAGE COMPRESSION

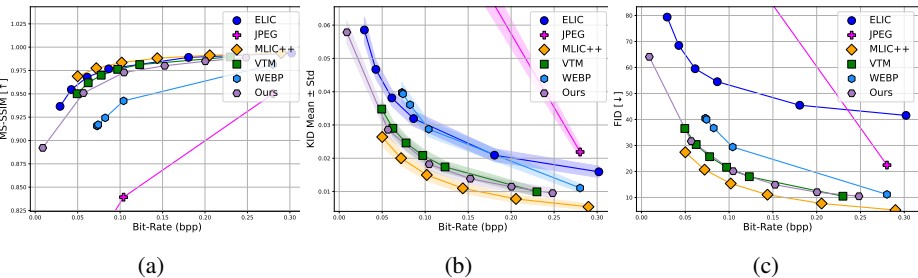

(a)            (b)            (c)

Figure 5: Comparison of performance-rate curves for image compression on the Taskonomy dataset using baseline methods and our proposed TAMC.

As shown in Fig. 5, TAMC demonstrates competitive performance compared to the VTM baseline. For structure- and texture-sensitive metrics, such as MS-SSIM (Wang et al., 2003), TAMC is slightly inferior to VTM. For semantic-sensitive metrics, including KID (Bińkowski et al., 2018), FID (Heusel et al., 2017) in Fig. 5, and LPIPS (Zhang et al., 2018) in Fig. 13, TAMC is slightly superior to VTM, reflecting its strong capacity for semantic understanding and perceptual quality retention. However, the Peak Signal-to-Noise Ratio (PSNR) results in Fig. 13a are comparatively lower, indicating room for improvement in pixel-level fidelity. This could be attributed to our choice of Xception (Chollet, 2017) as the encoder backbone and a task-specific decoder consisting of four transposed convolutional layers and four convolutional layers, which aligns with the architectures pre-trained for machine tasks (Fifty et al., 2021) but does not yet integrate the full advantages of advanced backbone modules for image compression, *e.g.*, GDN (Ballé et al., 2018), residual networks (Cheng et al., 2020), and transformers (Zou et al., 2022; Lu et al., 2022), suggesting potential for further exploration.

## 5 ANALYSIS

### 5.1 ABLATION OF GROUP INTER-COHERENT TASKS

| Method | Bitrate | Task Grouping1 | | | Task Grouping2 | | Task Grouping3 | | |
|---|---|---|---|---|---|---|---|---|---|
| | | Segment. | Depth | Normal | Normal | Keypoint | Segment. | Normal | Texture |
| **JPEG** | 0.283 | 44.32% | 29.86% | -7.05% | -14.10% | -44.58% | 28.40% | -17.56% | -0.06% |
| **WebP** | 0.281 | 3.13% | 20.08% | -3.69% | 2.79% | -49.00% | -0.91% | -5.43% | -0.59% |
| **VTM-17.0** | 0.229 | -0.24% | -5.28% | 8.24% | 11.16% | -46.65% | -1.82% | 2.84% | -0.37% |
| **ELIC** | 0.302 | -1.50% | -0.03% | 6.48% | 9.69% | -39.11% | -2.47% | 2.27% | -0.90% |
| **MLIC++** | 0.289 | -1.18% | -1.05% | 6.65% | 9.79% | -40.40% | -2.26% | 2.47% | -0.93% |
| **TAMC** | 0.224 | -10.08% | 3.33% | 0.21% | 12.43% | -45.02% | -4.84% | 5.38% | -1.05% |

Table 1: Performance loss reduction of grouped inter-coherent tasks relative to single-task training.

We also conducted ablation to evaluate the effectiveness and generalization of task grouping in the context of multi-task compression. Given 5 downstream tasks: {Semantic Segmentation, Depth Z-buffer, Edge Texture, Surface Normal, Keypoints 2D}, our task grouping results based on gradient coherence between task pairs are as follows: **Group 1** {Semantic Segmentation, Depth Z-buffer, Surface Normal}, **Group 2** {Surface Normal, Keypoints 2D} and **Group 3** {Semantic Segmentation, Surface Normal, Edge Texture. During inference, only the tasks marked in green within each group are decoded and referenced for downstream tasks, while the remaining tasks are solely used to enhance performance during training and discarded during inference.

For Compress-then-Analyze (CTA) methods, we assess the relative performance improvement of downstream tasks with vs. without task grouping when compression is performed before feature extraction. For our Analyze-then-Compress (ATC) paradigm, we evaluate the impact of task grouping in latent space, comparing performance with vs. without task grouping when feature extraction is conducted prior to compression. To isolate the effect of task grouping, the causal discovery module is not employed in this ablation study. The experimental results in Tab. 1 reveal that, for most baseline

methods, the performance of tasks such as Semantic Segmentation in Group 1, Keypoints 2D in Group 2, and Edge Texture in Group 3 exhibits loss reductions compared to single-task training, benefiting from information shared with other tasks. Depth Z-buffer and Surface Normal in Group 1, while not demonstrating significant improvements and even experiencing slight performance declines, contribute positively to the performance enhancement of other tasks within their respective groups. JPEG and WebP suffer the most in Semantic Segmentation and Depth Z-buffer, likely due to pixel-level distortions that introduce deviations in feature distributions, ultimately leading to degraded performance in task grouping. TAMC does not significantly degrade performance in any task, making it the most stable method. Furthermore, TAMC outperforms all other methods, demonstrating strong generalization across different task groups.

## 5.2    ABLATION OF CAUSAL DISCOVERY TOPOLOGY

To better understand how causal discovery graphs constructed via conditional entropy optimize both collaborative bitrate and task performance, we analyzed their effectiveness in determining the optimal trade-off between these factors. As visualized in Fig. 6, different graph construction methods are compared across various settings. In Setting A, where the graph follows principles of causal discovery via conditional entropy, the resulting tasks exhibit better coordination, balancing both bitrate efficiency and task performance. In contrast, Setting B and Setting C represent random graph constructions that do not adhere to causal discovery principles. The visualized results show that Setting B requires a higher bitrate to achieve similar performance while Setting C not only consumes more bitrate but also results in performance degradation. This comparison highlights the importance of proper graph construction in multi-task compression systems.

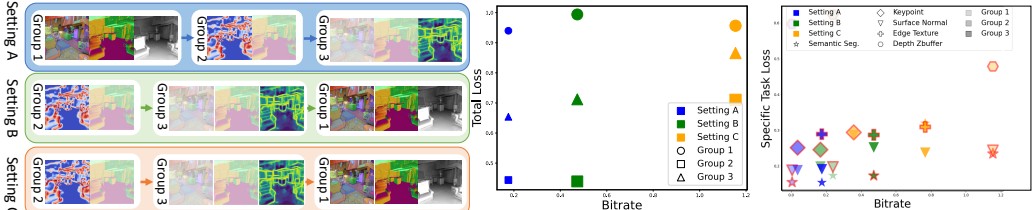

Figure 6: Impact of different DAG topologies on bitrate and multi-task performance (lower left indicates better solutions, where both bitrate and performance loss are minimized). **Left**: Example of graph construction methods. **Middle**: Task performance of graph node and bitrate consumption of the entire causal graph. Note: Different colors denote different graph topologies, and different shapes represent different nodes. **Right**: A more detailed breakdown of the middle, showing bitrate consumption and performance for different nodes in each causal graph. Note: Different colors represent different topologies, with varying transparency of the same color indicating different nodes within the same graph. Different shapes represent different tasks. Red-bordered shapes denote the best-performing task in each topology (prioritizing task performance over bitrate).

## 6    CONCLUSION

In this work, we introduced a novel multi-task representation compression framework that leverages causal discovery via conditional entropy to optimize the trade-off between bitrate efficiency and task performance. By grouping mutually beneficial tasks and constructing a DAG to characterize their interdependencies, our method enables efficient compression of disentangled representations. Through extensive experiments on key computer vision tasks, we demonstrated the effectiveness of our approach in both bitrate reduction and task accuracy. Our findings highlight the importance of properly structured task groupings and causal relationships in multi-task compression, offering a promising direction for future work in video coding for machine learning and multi-task optimization.

### ACKNOWLEDGMENTS

This work was supported by the Program of Beijing Municipal Science and Technology Commission Foundation (No.Z241100003524010), in part by the National Natural Science Foundation of China under Grant 62088102, in part by AI Joint Lab of Future Urban Infrastructure sponsored by Fuzhou

Chengtou New Infrastructure Group and Boyun Vision Co. Ltd, and in part by the PKU-NTU Joint Research Institute (JRI) sponsored by a donation from the Ng Teng Fong Charitable Foundation.

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

# A APPENDIX

## A.1 DISCUSSION

While graph models can effectively capture complex relationships between multiple representation spaces, it is well-known that the computational complexity of adding or removing nodes in graph structures is high. This becomes especially challenging in open-set scenarios, where tasks or data points are continuously evolving. Developing efficient methods for dynamically adding and removing nodes while maintaining the integrity of the graph remains an important research question.

Another critical point is the representation compatibility between different tasks. Ideally, the representation form and model architecture should be customized to fit the specific requirements of each task and application scenario. Until a truly unified model emerges, representations and architectures might not always be fully compatible across tasks. In our current work, we used a shared model architecture and representations across tasks, which, while ideal for controlled experimentation, may not reflect the diversity seen in real-world applications. However, our experiments successfully validated the feasibility and effectiveness of using a causal graph model based on conditional entropy for multi-task compression under these controlled conditions.

## A.2 MORE DETAILS OF EXPERIMENTAL SETTINGS

We applied and measured 6 tasks in Taskonomy[4], which is listed below:

- **Semantic Segment:** The annotations include 18 unique labels, with 16 object classes, a "background" class, and an "uncertain" class. For this task, we evaluate compression performance at different compression rates using cross-entropy loss.

- **Keypoint 2D:** This task involves 2D keypoint heatmaps. We assess compression performance at different compression rates using the L1 loss.

- **Edge Texture:** This task involves detecting 2D edge textures. Similar to Keypoints2D, we evaluate performance using L1 loss at different compression rates.

- **Surface Normal:** This task includes surface normal images, centered at 127. To evaluate performance under different compression rates, we use the `rotate_loss`, which is commonly applied in image processing or volume rendering tasks. The loss computes the L1 difference between the output and target and compares the result across 9 different orientations to find the minimal loss. This ensures that the model's depth predictions remain consistent under rotational and translational transformations, which is crucial when dealing with real-world noise and variations.

- **Depth Z-buffer:** This task involves Z-buffer depth images, measured in units of 1/512m with a maximum range of 128m. Similar to the Normal task, we use `rotate_loss` at different compression rates, first calculating the L1 difference between the output and target, then comparing across 9 orientations to find the minimal loss.

- **Autoencoder:** This task reconstructs RGB images at a resolution of $512 \times 512$. We evaluate compression performance at different compression rates using the L1 loss. Additionally, we assess fidelity using PSNR/SSIM (Wang et al., 2003) and perceptual quality using LPIPS (Zhang et al., 2018) / KID (Bińkowski et al., 2018) / FID (Heusel et al., 2017).

For the compression baselines, we used the open source codes, *i.e.*, JPEG[5], WebP [6], VTM-17.0[7], ELIC [8], and MLIC++ [9]. For downstream tasks, we uniformly used the Taskonomy pre-trained [10] Xception encoder and task-specific decoder. TAMC directly performs coherent task grouping and

---

[4] `https://github.com/StanfordVL/taskonomy/tree/master/data` by (Zamir et al., 2018)
[5] `ftp://ftp.ijg.org/pub/jpeg/`
[6] `https://github.com/webmproject/libwebp`
[7] `https://vcgit.hhi.fraunhofer.de/jvet/VVCSoftware_VTM`
[8] `https://github.com/VincentChandelier/ELiC-ReImplemetation`
[9] `https://github.com/JiangWeibeta/MLIC`
[10] `https://drive.google.com/drive/folders/1XQVpv6Yyz5CRGNxetO0LTXuTvMS_w5R5?usp=sharing`

causal graph construction in the latent space, so its encoder and decoder follow the Xception structure of the pre-trained model. For the image compression task, better model architecture designs are already available to optimize performance.

### A.3 SUPPLEMENTARY ABLATION STUDIES

In this section, we extend our analysis to E2E compression involving multiple supervised tasks as auxiliary tasks. We also examine single-task groups, where each task is treated as an independent group rather than grouping tasks together. Additionally, we explore whether the conherence from Task A to Task B can be used to predict the conherence from Task B to Task A. Furthermore, we investigate the impact of prohibiting the same task from appearing in multiple clusters and assess whether this restriction leads to better or worse performance. Finally, we compare our approach with VQ-GAN-based compression method to evaluate its overall effectiveness.

#### A.3.1 ADDITIONAL RESULTS OF E2E COMPRESSION WITH MULTIPLE TASK AUXILIARY LOSS

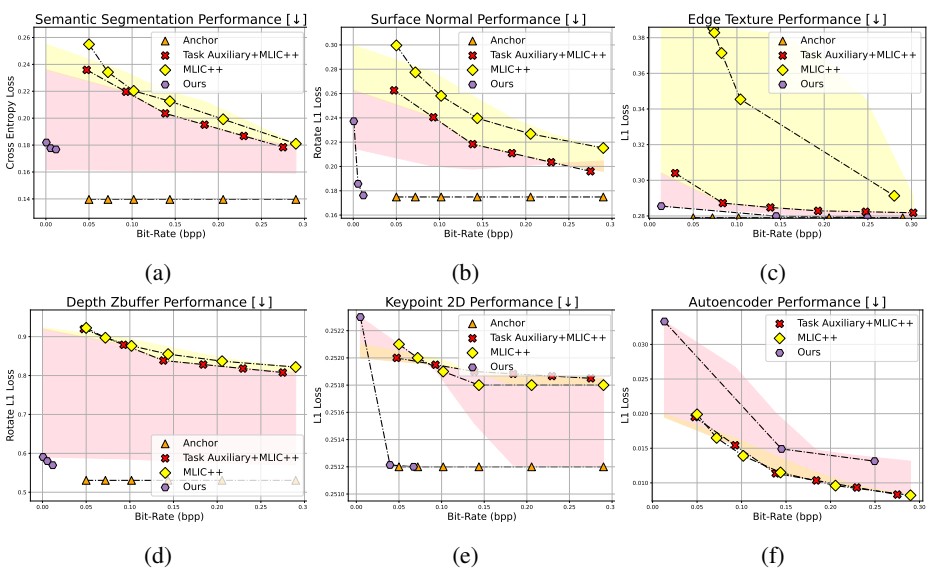

Figure 8: Performance-rate curves for 6 tasks: Semantic Segmentation, Depth Z-buffer, Surface Normal, Keypoint 2D, Edge Texture, and Autoencoder, on the Taskonomy dataset. The comparison includes baseline compression methods ( MLIC++, Task Auxiliary+MLIC++) and our proposed TAMC. Results highlight the efficiency of our method in achieving superior task performance at various bit rates, demonstrating the necessity of task grouping and scalable encoding.

To evaluate the impact of task grouping and the DAG on performance, we conducted an ablation study where both components were removed. In this alternative setup, multiple auxiliary tasks were integrated directly into the compression framework, as illustrated in Fig. 7, optimizing the following combined loss function:

$$\mathcal{L} = \mathcal{L}_{\text{compression}} + \sum_i w_i . \mathcal{L}_{\text{task}_i} \qquad (14)$$

Here, $w_i$ represents the weight assigned to each task, we set equal weights for tasks in our ablation studies. We tested this approach on 6 tasks: Semantic Segmentation, Surface Normal, Edge Texture, Depth Z-buffer, Keypoint 2D, and Autoencoder. Fig. 8 presents the performance-rate

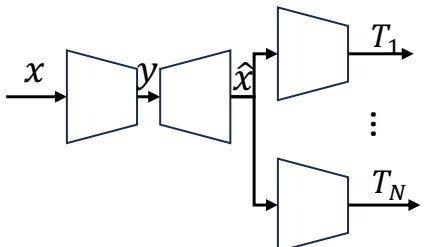

Figure 7: End-to-end compression with Multiple Task Auxiliary Loss.

curves for each task, comparing our method to baseline compression methods (e.g., MLIC++) and an auxiliary task-based variant (Task Auxiliary+MLIC++). The results demonstrate the advantages of

our approach in both compression efficiency and task performance, as well as the complex interplay between task collaboration and conflict, which highlights the significance of task grouping and scalable encoding.

Our method consistently outperforms baseline compression methods, especially for tasks such as **Semantic Segmentation**, **Surface Normal**, **Edge Texture**, and **Depth Z-buffer**. Even at lower bit rates, our framework achieves notable improvements over MLIC++ and Task Auxiliary+MLIC++, underscoring its robustness and adaptability in multi-task settings. These results confirm the efficacy of task grouping and the integration of a causal DAG in preserving task performance under constrained compression conditions.

The Task Auxiliary+MLIC++ variant, which replaces task grouping and DAG with auxiliary tasks, provides useful insights into task-level interactions. For tasks like **Edge Texture**, **Semantic Segmentation**, **Surface Normal**, and **Depth Z-buffer**, the auxiliary task approach yields substantial improvements compared to end-to-end compression methods, suggesting enhanced task collaboration and feature sharing. However, for **Autoencoder**, the auxiliary task approach performs similarly to end-to-end methods, indicating limited benefits for tasks with strong self-supervised structures.

In contrast, the **Keypoint 2D** task experiences performance degradation with the auxiliary task approach, likely due to task interference. This highlights the potential conflicts between task objectives, emphasizing the importance of careful task grouping to mitigate such issues.

The observed interplay of task collaboration and conflict further validates the need for task grouping. By grouping tasks based on gradient coherence, our framework minimizes inter-task interference and promotes effective task collaboration, explaining its superior performance relative to the auxiliary task-based approach. Moreover, these results show that uncoordinated task interactions can negatively impact specific tasks, such as **Keypoint 2D**.

Fig. 9 also reveals variations in bit-rate efficiency across tasks. For instance, **Semantic Segmentation** and **Surface Normal** maintain strong performance even at lower bit rates, while tasks like **Edge Texture** require higher bit rates due to the need for detailed feature representation. These findings highlight the importance of scalable encoding to accommodate the varying bit-rate needs of different tasks. By enabling task grouping and scalable encoding, our method addresses these challenges while optimizing compression efficiency.

### A.3.2 PERFORMANCE COMPARISON OF SINGLE TASK VS. GROUPED TASKS

| Method | Semantic Seg. | | Depth Z-buffer | | Surface Normal | | Keypoint 2D | |
|---|---|---|---|---|---|---|---|---|
| | Bitrate | Test Loss | Bitrate | Test Loss | Bitrate | Test Loss | Bitrate | Test Loss |
| Single Task | 0.0014 | 0.0852 | 0.0008 | 0.2925 | 0.0006 | 0.1315 | 0.0017 | 0.2439 |
| | 0.0055 | 0.0680 | 0.0051 | 0.2648 | 0.0052 | 0.0963 | 0.0645 | 0.1115 |
| | 0.0069 | 0.0674 | 0.0063 | 0.2643 | 0.0069 | 0.0938 | 0.0940 | 0.0954 |
| Group 1 | 0.0015 | 0.0704 | 0.0015 | 0.2615 | 0.0015 | 0.1378 | - | - |
| | 0.0096 | 0.0598 | 0.0096 | 0.2419 | 0.0096 | 0.1079 | - | - |
| | 0.0139 | 0.0574 | 0.0139 | 0.2385 | 0.0139 | 0.1045 | - | - |
| Group 2 | - | - | - | - | 0.0018 | 0.1528 | 0.0018 | 0.2412 |
| | - | - | - | - | 0.0623 | 0.1103 | 0.0623 | 0.0944 |
| | - | - | - | - | 0.0843 | 0.1080 | 0.0843 | 0.0936 |

Table 2: Performance-Bitrate comparing task grouping with single task. Group 1 represents the task grouping of Semantic Segmentation, Depth Z-buffer, and Surface Normal. Group 2 represents the task grouping of Surface Normal and Keypoint 2D.

To further examine the effectiveness of task groups, we trained a model based on single-task compression, where each task is treated as an independent group rather than grouping tasks together. Additionally, we set up a Grouping 1 compression model that jointly optimizes the task grouping of Semantic Segmentation, Depth Z-buffer, and Surface Normal, and a Group 2 compression model that jointly optimizes the task grouping of Surface Normal and Keypoint 2D. The results are shown in Tab.2, and further visualized in the performance curves in Fig. 9. As observed, Semantic Segmentation and Depth Z-buffer benefit from a more compact and high-precision representation when

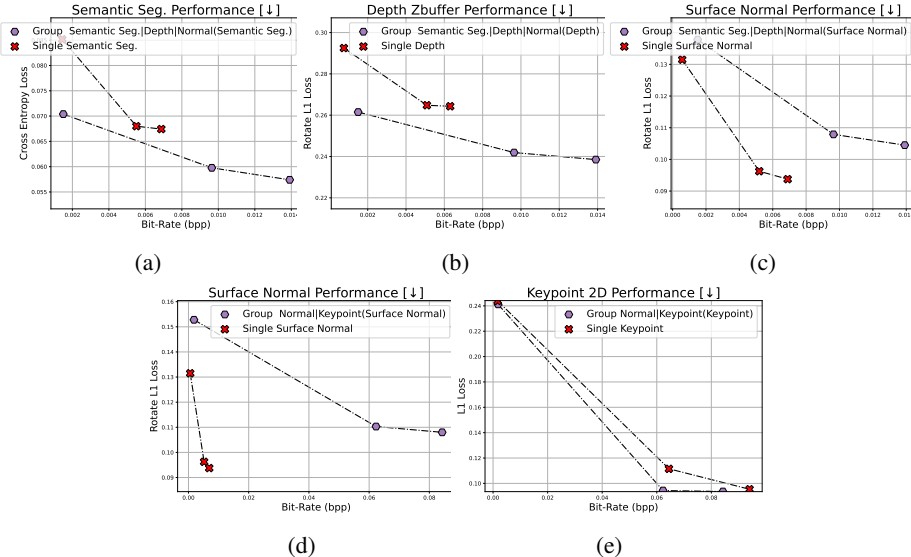

Figure 9: Performance-Bitrate curves comparing task grouping with single task. The purple curves represent the bitrate-performance performance of task grouping, while the red curves represent the performance of the single-task model.

| Method | | Semantic Seg. | Depth Z-buffer | Surface Normal | Edge Texture | Keypoint 2D | Total Test Loss |
|---|---|---|---|---|---|---|---|
| Setting 1 | Group 1 | **0.0535** | **0.2515** | **0.1070** | — | — | 0.5279 |
| | Group 2 | — | — | — | **0.0234** | **0.0925** | |
| Setting 2 | Group 1 | **0.0535** | **0.2515** | **0.1070** | — | — | 0.5223 |
| | Group 2 | — | — | 0.1086 | **0.0274** | **0.0743** | |

Table 3: Number of Task Grouping=2. Tasks are not allowed to appear in multiple groups in Setting 1. Tasks can appear in multiple groups in Setting 2.

| Method | | Semantic Seg. | Depth Z-buffer | Surface Normal | Edge Texture | Keypoint 2D | Total Test Loss |
|---|---|---|---|---|---|---|---|
| Setting 1 | Group 1 | **0.0531** | **0.2624** | — | — | — | 0.5298 |
| | Group 2 | — | — | — | **0.0234** | **0.0925** | |
| | Group 3 | — | — | **0.0984** | — | — | |
| Setting 2 | Group 1 | 0.0535 | **0.2515** | 0.1070 | — | — | 0.4866 |
| | Group 2 | **0.0495** | — | **0.1034** | **0.0244** | — | |
| | Group 3 | — | — | 0.1100 | — | **0.0573** | |

Table 4: Number of Task Groupings=3.

grouped in Group 1. Similarly, Keypoint 2D shows improved representation with higher compactness and precision in Group 2. Surface Normal's performance remains comparable between task grouping and the single-task approach.

One additional benefit of task grouping is the shared encoder and shared representations across multiple tasks. For example, in Group 1, a single encoder feature extraction is used for inference, consuming 0.0015 bpp for Semantic Segmentation, Depth Z-buffer, and Surface Normal. In contrast, treating each task independently requires three separate feature extraction inferences, with a total bitrate of 0.0014 bpp + 0.0008 bpp + 0.0006 bpp = 0.0028 bpp to serve the three tasks.

### A.3.3 IMPACT OF TASK EXCLUSIVITY ACROSS GROUPS ON PERFORMANCE

From a performance ceiling perspective, allowing the same task to appear in different clusters maximizes the potential for task collaboration. As shown in Tab. 3 and 4, we conducted experiments under two settings: Setting 1, where tasks are not allowed to appear in multiple clusters, and Setting

2, where tasks can appear in multiple clusters. Setting 2 achieves a superior performance ceiling, demonstrating the advantages of task interdependence. Notably, since each task is only inferred once, Setting 2 does not introduce additional inference complexity. However, it does result in a significant increase in GPU memory consumption during training. This highlights the trade-off between performance and resource utilization when task exclusivity is relaxed.

### A.3.4 IMPACT OF TASK ORDER ON TASK GROUPING AND ADDRESSING VARIABILITY

Regarding whether the order of tasks affects the cost calculation and how to address potential variability in the method using coherence scores: In Sec. 3.2, the number of possible groupings for $n$ tasks is given by the Bell numbers. To quickly estimate the similarity between tasks, after calculating $\hat{C}_{u \to v}$, $\hat{C}_{w \to v}$, we estimate the higher-order costs for $\{\tau_u, \tau_v, \tau_w\}$ which significantly reduces the computational complexity. We mitigate the impact of task order with the following operations:

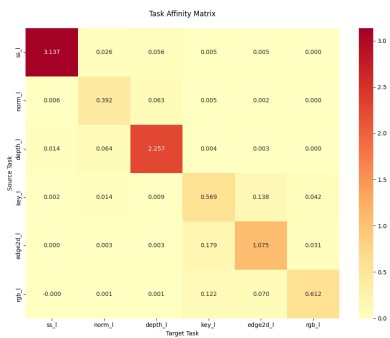

Figure 10: Inter-task coherence on Taskonomy. Red color signify higher inter-task affinities.

1. The coherence score is a measure based on the impact of gradient updates between tasks on the loss function. It is a relative measure that reduces the impact of the absolute order of task execution. 2. By calculating the coherence scores throughout the entire training process and taking the average, we can mitigate the impact caused by specific stages of training, thus reducing the potential variability brought about by changes in the order of tasks. 3. To validate the reasonableness of this operation, in Fig.10, we experimentally demonstrate that although the coherence score between task pairs is not strictly symmetric ($\hat{C}_{u \to v} \neq \hat{C}_{v \to u}$), it exhibits a strong symmetry trend in practice. This allows us to approximate the values while maintaining accuracy and efficiency.

### A.3.5 COMPARISON WITH VQ-GAN COMPRESSION

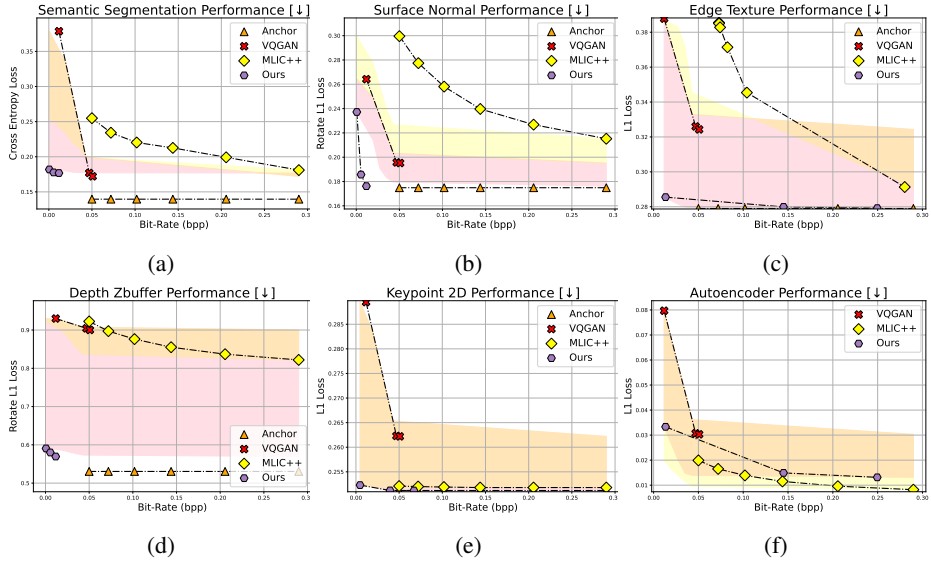

Figure 11: Comparison with VQGAN (Mao et al., 2024) in multi-task compression: Our method outperforms VQGAN in compression efficiency at low bitrates and closely matches the supervision anchor at higher bitrates, particularly in tasks like Keypoint 2D, Semantic Segmentation, and Depth Z-buffer. VQGAN, however, shows performance degradation in fine-grained tasks, highlighting the advantages of our task-aware semantic compression approach.

While VQGAN-based compression methods (Esser et al., 2021; Mao et al., 2024) achieve perceptual compression at low bitrates through discretized codebooks, our optimization goal diverges significantly. Unlike VQGAN (Mao et al., 2024), which prioritizes image reconstruction and perceptual quality, our approach focuses on compact, multi-task semantic compression. Specifically, we optimize for efficient semantic representation sharing across tasks, reducing redundancy in encoding. This contrasts with VQGAN's generative approach. Comparative experiments in Fig. 11 reveal key differences in performance and efficiency.

Although VQGAN (Mao et al., 2024) demonstrates significant improvements in compression rates over VAE backbone models, especially for perceptual tasks, it still introduces bias compared to the optimal supervision anchor, even with sufficient bitrate. This is due to its generative nature, which cannot fully eliminate reconstruction errors, leading to discrepancies from the ground truth. Moreover, VQGAN struggles with tasks requiring precise, sparse local feature detection (*e.g.*, Keypoint 2D), a limitation common in generative models that fail to capture fine-scale features. In contrast, our method is specifically designed to encode and preserve task-specific features, achieving superior performance in these tasks.

In multi-task settings (*e.g.*, Semantic Segmentation, Surface Normal, Edge Texture, Depth Z-buffer), our approach demonstrates superior compression efficiency at low bitrates and near-optimal performance at higher bitrates, closely matching the supervision anchor. VQGAN, while effective for perceptual compression, struggles to leverage task-specific semantic information, leading to inferior performance, particularly at low bitrates.

### A.4 COMPLEXITY ANALYSIS

#### A.4.1 COMPLEXITY ANALYSIS OF COMPRESSION FOR MULTIPLE TASKS: ATC VS. CTA PARADIGMS

| MLIC++ Encoder/Decoder Module | $g_a$ | $g_s$ |
|---|---|---|
| KParams | 12033.6 | 4396.3 |
| MMACs | 194556.9 | 296377.9 |

Table 5: Parameters and Forward Macs of Encoder/Decoder of MLIC++ on $512 \times 512$ images.

| Task Xception Encoder/Decoder Module | $l_a$ | $l_s$ |
|---|---|---|
| KParams | 16467.2 | 525.1 |
| MMACs | 25708.0 | 4968.1 |

Table 6: Parameters and Forward Macs of Task Encoder/Decoder of Xception on $512 \times 512$ images.

In the task of compression for multiple downstream tasks, we investigate two compression paradigms: Analysis and Then Compression (ATC) and Compression and Then Analysis (CTA). ATC pipeline consists of two main phases: the downstream task analysis phase, where the input image is used for specific tasks such as keypoint detection, segmentation, and depth estimation, and the feature compression phase, which includes the encoding and decoding steps. The CTA pipeline also consists of two main phases: the image compression phase, which includes encoding and decoding, and the downstream task analysis phase, where the compressed image is used for tasks such as keypoint detection, segmentation, and depth estimation. To better understand the computational costs involved in each module, we summarize the parameters and forward MACs (Multiply-Accumulate Operations) of the different components in the ATC and CTA pipelines. These values are presented in the tables 5, 6, 7. Below, we present the complexity analysis for both approaches.

Our proposed method belongs to ATC paradigm, and the total computational complexity of this pipeline results from both the downstream task analysis and the feature compression phases. The number of downstream tasks $N$ and the number of task groups $K$ directly affect the computational cost of the task analysis phase. The overall complexity is expressed as:

$$\text{Total Complexity of ATC} = N \cdot l_s + K \cdot (l_a + \boldsymbol{\theta}_{hd} + \boldsymbol{\theta}_{cm+} + \boldsymbol{\theta}_{gc})$$
$$= N \cdot 4968.1 + K \cdot 40088.0, \tag{15}$$

| Context Module | $\boldsymbol{\theta}_{hd}$ | $\boldsymbol{\theta}_{cm}$ | $\boldsymbol{\theta}_{cm+}$ | $\boldsymbol{\theta}_{gc}$ |
|---|---|---|---|---|
| KParams | 5810.1 | 755.2 | 1487.7 | 2264.4 |
| MMACs | 8925.1 | 1148.2 | 2264.4 | 7100.5 |

Table 7: Parameters and Forward Macs of entropy context modules on $512 \times 512$ images.

where $\boldsymbol{\theta}_{hd}$, $\boldsymbol{\theta}_{cm+}$, and $\boldsymbol{\theta}_{gc}$ represent the complexities of the context modules. $N$ is the number of downstream tasks, $K$ is the number of task groups, $l_a$ and $l_s$ represent the complexities of the Task Xception Encoder/Decoder for each individual task.

This formulation indicates that the task analysis phase (which involves both $l_a$ and $l_s$) and the context module complexities contribute to the total computational cost.

The total computational complexity in the CTA pipeline arises from both the compression and the downstream task analysis phases. The number of tasks $N$ directly affects the computational cost of the task analysis phase. The overall complexity is expressed as:

$$
\begin{aligned}
\text{Total Complexity of CTA} &= g_a + g_s + \boldsymbol{\theta}_{hd} + \boldsymbol{\theta}_{cm} + \boldsymbol{\theta}_{gc} + N \cdot (l_a + l_s) \\
&= 500008.6 + N \cdot 30676.1,
\end{aligned}
\tag{16}
$$

where $g_a$ and $g_s$ are the complexities of the MLIC++ Encoder/Decoder. $\boldsymbol{\theta}_{hd}$, $\boldsymbol{\theta}_{cm}$, and $\boldsymbol{\theta}_{gc}$ represent the complexities of the context modules. $N$ is the number of downstream tasks. $l_a$ and $l_s$ represent the complexities of the Task Xception Encoder/Decoder for each individual task.

This formulation emphasizes the contribution of the encoding/decoding processes (represented by $g_a$ and $g_s$) as well as the task-specific encoding/decoding complexities $l_a$ and $l_s$ in the CTA pipeline.

### A.4.2 COMPLEXITY ANALYSIS OF OUR LOOKAHEAD MODULE

Our Lookahead module consists of two steps: task grouping and DAG construction. In the task grouping step, we first compute the coherence score between tasks using joint training on $N \times N$ task pairs. Then, we use pairwise coherence scores to estimate the coherence scores of higher-order task groupings. Finally, based on all task groups, task coherence scores and budget $b$, we select $k$ multitask networks to maximize the overall task performance. This is an NP-hard problem. A brute force approach would take $O(|\mathcal{T}| \cdot |C_0|^{\frac{b}{\min_{n \in C_0} c_n}})$, which is exponential in the maximum number of groups that fit within the budget. Here, $|\mathcal{T}|$ is the number of tasks, $|C_0|$ is the number of candidate networks, $b$ is the budget, and $\min_{n \in C_0} c_n$ is the smallest cost among the networks. This can be solved using the branch-and-bound-like algorithm provided in prior work (Zamir et al., 2018) as detailed in Sec.A.5.

---

**Algorithm 2** Get Task Grouping Strategy

---

**Input:** $C_r$, a running set of candidate groups, each with an associated cost $c \in \mathbb{R}$ and a performance score for each task the network solves. Initially, $C_r = C_0$
**Input:** $S_r \subseteq C_0$, a running solution, initially
**Input:** $b_r \in \mathbb{R}$, the remaining budget, initially $b$

  1: **function** GETBESTNETWORKS($C_r, S_r, b_r$)
  2:      $C_r \leftarrow$ FILTER($C_r, S_r, b_r$)
  3:      $C_r \leftarrow$ SORT($C_r$)                                  ▷ Most promising groups first
  4:      $Best \leftarrow S_r$
  5:      **for** $n \in C_r$ **do**
  6:          $C_r \leftarrow C_r \setminus n$                              ▷ $\setminus$ is set subtraction.
  7:          $S_i \leftarrow S_r \cup \{n\}$
  8:          $b_i \leftarrow b_r - c_n$
  9:          $Child \leftarrow$ GETBESTNETWORKS($C_r, S_i, b_i$)
10:          $Best \leftarrow$ BETTER($Best, Child$)
11:      **end for**
12:      **return** $Best$
13: **end function**

14: **function** FILTER($C_r, S_r, b_r$)
15:      Remove groups from $C_r$ with $c_n > b_r$.
16:      Remove groups from $C_r$ that cannot improve $S_r$'s performance on any task.
17:      **return** $C_r$
18: **end function**

19: **function** BETTER($S_1, S_2$)
20:      **if** $C(S_1) < C(S_2)$ **then**
21:          **return** $S_1$
22:      **else**
23:          **return** $S_2$
24:      **end if**
25: **end function**

---

The overall time complexity of the DAG construction algorithm 1 is dominated by the nested loops that compute conditional and independent entropies for each latent group. The algorithm iterates over all $K$ groups, leading to an outer loop complexity of $O(K)$. For each group, the algorithm computes the conditional entropy between pairs of latent variables. This operation takes $O(n)$ time, where $n$ is the size of the dataset (number of samples). Additionally, the algorithm computes the independent entropy for each group, which takes $O(m)$ time, where $m$ represents the complexity of entropy calculation for a single group Thus, the overall time complexity is:

$$O(K^2 \cdot n) + O(K \cdot m),$$

where $K$ is the number of groups, $n$ is the time complexity for calculating conditional entropy, $m$ is the time complexity for calculating independent entropy for a group.

The process of computing task coherence scores, grouping tasks, and constructing the corresponding Directed Acyclic Graph (DAG) is computationally intensive. These steps share a conceptual similarity with the lookahead stage in traditional video encoding (Li, 2003; He & Mitra, 2002; Wang & Kwong, 2008; Ma et al., 2005). In video encoding, the lookahead stage performs a preliminary analysis of the video content to optimize the encoding process, ensuring that the final compression achieves the best trade-off between quality and bitrate. Specifically, it evaluates potential encoding decisions ahead of time to minimize redundancies and improve efficiency, all while adhering to bitrate constraints.

Similarly, in multi-task compression, the task grouping and causal relationship modeling steps aim to optimize the encoding of multiple tasks by leveraging the inherent interdependencies among them. However, due to the high computational complexity of calculating coherence scores between tasks and determining the optimal task groupings, an efficient pre-analysis phase is essential. Our contribution lies in exploring the effectiveness of task grouping and cross-task causal relationship

modeling, demonstrating that these techniques can significantly enhance the performance of multi-task compression.

To make this process feasible and scalable, further exploration can involve using downsampled low-resolution images as inputs for the pre-analysis phase. This strategy, when combined with a low-complexity feature extraction backbone (Iandola, 2016), provides an efficient means of assessing task grouping performance under bitrate consumption constraints.

## A.5 IMPLEMENTATION DETAILS OF TASK GROUPING

As mentioned in Sec. 3.2, we adopt a branch-and-bound method, as in prior works (Standley et al., 2020; Zamir et al., 2018), to search for locally optimal grouping strategies under the given bitrate budget $b$. Here we provide a pseudo-algorithm for clarity, and the implementation is provided in Github[11]. Consider the situation in which we have an initial candidate set $C_0 = \{n_1, n_2, ..., n_m\}$ of fully trained networks that each solve some subset of our task set $\mathcal{T}$. Our goal is to choose a subset of $C_0$ that solves all tasks in budget $b$ and the lowest overall loss. More formally, we want to find a solution $S_b = \arg\min_{S \subseteq C_0 : \text{cost}(S) \leq b} \mathcal{L}(S)$. It can be shown that solving this problem is NP-hard in general (reduction from SET-COVER). A brute-force approach would take $O(|\mathcal{T}| \cdot |C_0|^{\frac{b}{\min_{n \in C_0} c_n}})$, which is exponential in the maximum number of groups that fit in our budget. This would be computationally challenging even for small problems.

However, many techniques exist that can optimally solve *most* instances of problems like these in reasonable amounts of time. All of these techniques produce solutions that perform equally well. We chose to use a branch-and-bound-like algorithm for finding this optimal solution (shown in Algorithm 2), but in principle the same solution could be achieved by other optimization methods, such as encoding the problem as a binary integer program (BIP) and solving it in a way similar to Taskonomy(Zamir et al., 2018). Algorithm 2 chooses the best subset of groups in our collection, subject to the inference time budget constraint. The algorithm recursively explores the space of solutions and prunes branches that cannot lead to optimal solutions. The recursion terminates when the budget is exhausted, at which point $C_r$ becomes empty and the loop body does not execute. The sorting step on line 3 requires a heuristic upon which to sort. We found that ranking models based on how much they improve the current solution, $S$, works well. It should be noted that this algorithm always produces an optimal solution, regardless of which sorting heuristic is used. However, better sorting heuristics reduce the running time because subsequent iterations will more readily detect and prune portions of the search space that cannot contain an optimal solution.

## A.6 MORE EXPERIMENTAL RESULTS

In Fig. 12, we present examples showing competitive qualitative results from our method compared to VTM and end-to-end compression methods. Fig. 13 provides a detailed comparison of PSNR-Bitrate and LPIPS-Bitrate performance.

---

[11]https://github.com/tstandley/taskgrouping/

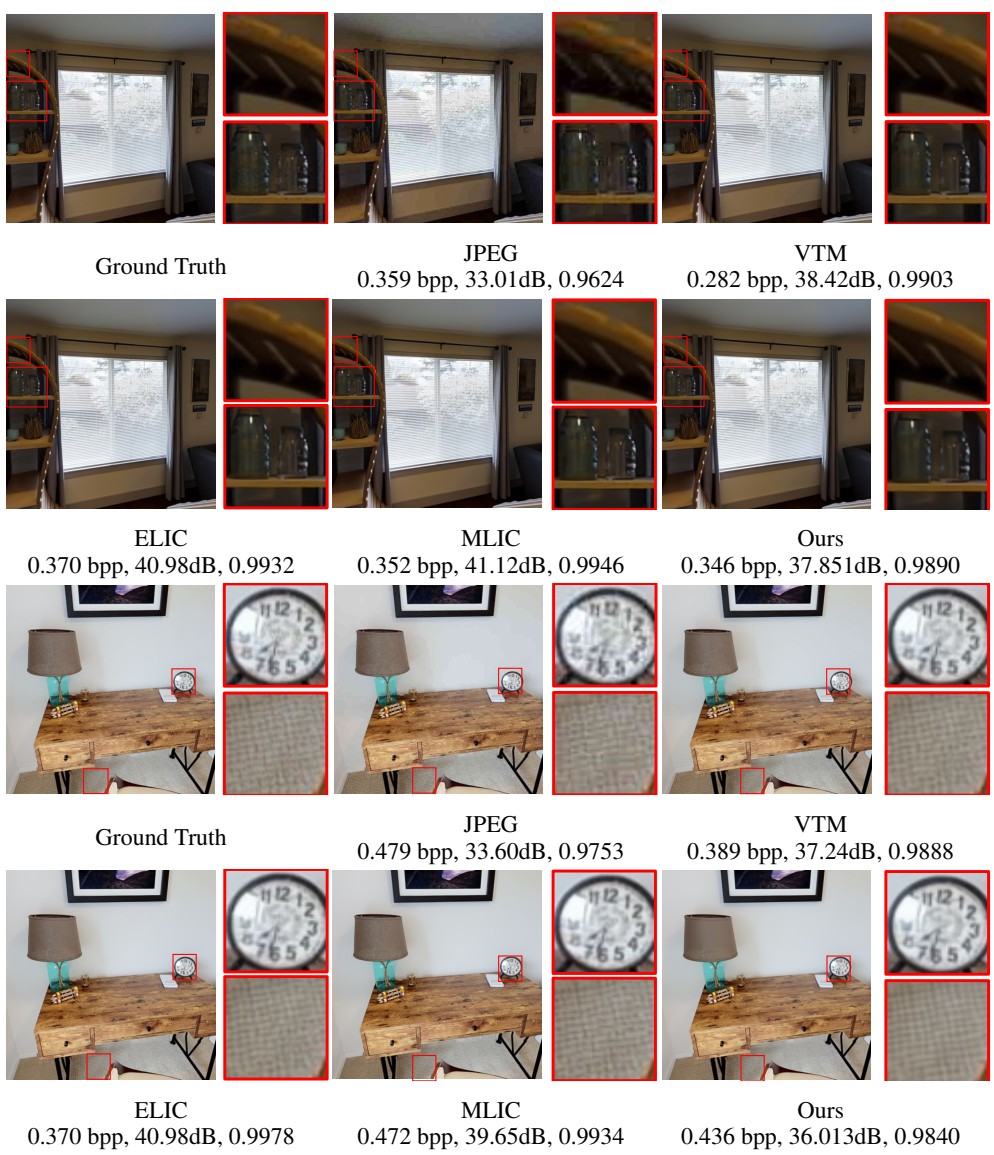

Figure 12: Subjective comparisons.

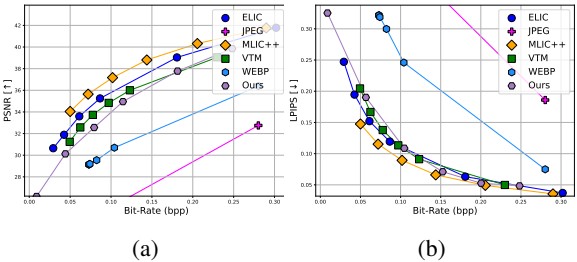

(a)                                        (b)

Figure 13: Comparison of PSNR-Bitrate performance and LPIPS (Zhang et al., 2018)-Bitrate performance for image compression on the Taskonomy dataset, using baseline methods and our proposed TAMC.

