# OpenReview forum: "Which Tasks Should Be Compressed Together? A Causal Discovery Approach for Efficient Multi-Task Representation Compression"
_ICLR.cc/2025/Conference — ICLR 2025 Poster_

### Official Review · Reviewer_5H7f · 2024-10-24

**Soundness:** 2
**Presentation:** 1
**Contribution:** 3
**Rating:** 3
**Confidence:** 4

**Summary:**

In this paper, the authors propose a data compression technique using representations that can be applied to multi-tasking in computer vision. They propose a methodology for task-aware representation learning by using multi-task learning approaches and grouping similar tasks based on the alignment of gradient vectors during the training phase. To efficiently utilize the learned representations, they calculate the causal relationships between task groups using conditional entropy and construct a directed acyclic graph (DAG). The proposed method was validated using the Taskonomy dataset, and the baselines included traditional data compression methods (e.g., JPEG, WEBP) as well as some recent methods (e.g., ELIC, MLIC++, etc.). The authors confirmed the robustness of the proposed method at relatively lower bit-rates compared to the baselines across multiple tasks.

**Strengths:**

- Hierarchical structure of multiple tasks: The authors propose a methodology that learns a universal representation for computer vision tasks by utilizing techniques from multi-task learning and uses these representations to form a hierarchical structure of the tasks. I believe this methodology could aid research on general representations in areas beyond data compression.
- Robustness of the trained representation: At lower bit-rates, the performance on downstream tasks consistently outperforms trained compression methods (such as MLIC++, ELIC).
- Ablation with random graphs: The authors compared the performance of the proposed causal DAG structure, which exploits causal relationships, with that of a random graph structure to verify its effectiveness. This demonstrates the existence of inter-task relationships and shows that efficient learning and inference, which take these relationships into account, have a considerable impact on performance.

**Weaknesses:**

- Lack of clarity, not well-organized components: The figures in the paper are generally not well-organized, and the font size within the figures is excessively small, causing visibility issues. In particular, in Figure 2, the font size within the figure is less than half the line height. Even if it results in a reduction of the amount of information in each figure, it seems necessary to increase visibility by focusing on the most important information.
- Relatively low data decoding accuracy: As mentioned by the authors, the results in subsection 4.3.2 show that the proposed method performs similarly or relatively worse than existing methods in terms of image compression. In order to show that this difference is not a significant problem for the application of the method, it seems necessary to qualitatively compare the decoded images, at least with some examples.
- No results with time and space complexity: One of the critical aspects of a data compression algorithm is minimizing the time and space cost of its execution. Since this work focuses on data compression techniques using multi-task learning methods rather than representation learning methods, I believe it is necessary to analyze such costs to justify the applicability of the proposed method.
- Lack of baselines for comparison: While I think it seems meaningful that the paper uses traditional data compression techniques (e.g., JPEG, WEBP) as baselines, there is still a lack of comparison with more recent trainable methods. Even if the compared baselines are considered state-of-the-art models, I believe that additional comparisons are necessary to clarify the consistent robustness of the proposed method compared to other approaches.
- To summarize, the paper lacks completeness in explaining the methodology and convincing the readers through the results, and the experiments are not sufficiently thorough. Regarding the experiments, it may be helpful for the authors to refer to the analytical techniques used in the MLIC++ paper [1], which they have cited as a key reference. Through a comprehensive revision of the content and additional analyses, the paper’s clarity can be improved, and its novelty can be further emphasized.

[1] Wei Jiang et al. MLIC++: Linear Complexity Multi-Reference Entropy Modeling for Learned Image Compression. https://arxiv.org/abs/2307.15421

**Questions:**

- Although the paper states that tasks were grouped based on gradient coherence, the justification for task grouping is not clearly stated in LN 470. Could the task grouping process be further explained?
- In Appendix A.5.2, the authors state that they select the parent node for a chosen child based on the node with the highest mutual information and claim that this is equivalent to the optimal choice for conditional entropy. While this claim seems correct, in the partial ordering of Algorithm 5, the child node is not explicitly defined, so the claim does not appear to be directly related to the implementation of the proposed method. It seems there might be an implicit assumption that the task with the larger entropy in its representation distribution is assigned the parent node. Would this be an accurate interpretation?
- It is mentioned that the total bit-rate of task-wise representations was used as a regularization for model training, but there is not enough explanation regarding the bit-rate for each task in the trained models. It would be helpful to include information on the observed bit-rates for each task and to explain how these bit-rates may be related to the tasks.
- Although theoretical approaches are mentioned in Appendix A.5.4 and A.5.5, they do not seem to be fully utilized or proven within the logic of the paper. Is there a plan to elaborate on these? In addition, could further explanations be provided in relation to the above Weaknesses and Questions?
- (minor) Uniform upper and lower case conventions in subsection titles (especially in Section 5)
- (minor) Typo in Equation 3 (\theta_s -> \theta_s^t)
- (minor) Typos in Algorithm 1, 2 in Appendix (end for"\n"return)

---

> ### Author Response · Authors · 2024-12-01
> **Official Response to Reviewer 5H7f (Part 1/7)**
>
> Dear Reviewer  5H7f,
>
> Thank  you for the time and effort you have dedicated to review our work. We have carefully considered each of your comments and have addressed them in the revised version of the manuscript.
>
> ---
>
> **Q1: Lack of clarity, not well-organized components.**
>
> **A1:**  We have made several revisions aim to ensure that readers can more easily follow our research motivation, methodology, and implementation details.
>
> 1. **Improved font size and clarity:** We have carefully revised the font size and overall clarity in the figures. For instance, **Fig. 2** [Original main paper, L.162-184] has been updated to **Fig. 3** [revised main paper, L.162-182] with larger fonts to ensure better visibility and readability of key information.
>
> 2. **Visualization of task representations:**  In **Fig.1(c)** [revised main paper, L.37-42], we have added a visualization of the latent space across different tasks, which clearly demonstrates the redundancy, uniqueness, and collaboration of task representations. This helps emphasize our research motivation: **Task Grouping** can effectively leverage semantic collaboration, maximizing information sharing between related tasks.
> Additionally, **Causal Discovery via Conditional Entropy** models causal relationships between tasks, allowing child task representations to gain contextual information from the parent task layers, thereby reducing redundancy and uncertainty.
>
> 3. **Clarification of Taskonomy-Aware Compression:** In **Fig.2** [revised main paper, L.108-117] offers an overview of how we extend E2E Compression to Taskonomy-Aware Compression.
>
> 4. **Detailed Implementation of Taskonomy-Aware Compression:**  **Fig.4** [revised main paper, L.324-336] illustrated the parent task nodes provide local and global context that enhances the prediction accuracy of child task nodes, thus improving the model’s overall performance.
>
> By making these improvements, we believe the figures now better convey the essential concepts of our work.
>
>
> ---
>
> **Q2. Relatively low data decoding accuracy and add subjective results.**
>
> **A2:** Thank you for raising the concern regarding data decoding accuracy. In **Fig. 13** [revised appendix, L.1296-1335], we have provided subjective visual comparisons, where we present competitive qualitative results to VTM and end-to-end compression methods. These comparisons demonstrate that our method maintains commendable deocoding accuracy and  has concurrently enhanced  task accuracy.
>
> In terms of implementation, we adopted the backbone from prior work in Multi-Task Learning (MTL) [1,2,3], specifically using Xception [4] as the encoder backbone, and a relatively simple decoder composed of four transposed convolutional layers and four convolutional layers. However, this configuration does not yet fully exploit the capabilities of more advanced image compression architectures, such as GDN [5], residual networks [6], and transformers [7,8,9]. This leaves room for future optimization and enhancement.
>
> ---
>
> [1] Chris Fifty, Ehsan Amid, Zhe Zhao, Tianhe Yu, Rohan Anil, and Chelsea Finn. Efficiently identifying
> task groupings for multi-task learning. Advances in Neural Information Processing Systems, 34:
> 27503–27516, 2021.
>
> [2] Trevor Standley, Amir Zamir, Dawn Chen, Leonidas Guibas, Jitendra Malik, and Silvio Savarese.
> Which tasks should be learned together in multi-task learning? In International conference on
> machine learning, pp. 9120–9132. PMLR, 2020.
>
> [3] Amir R Zamir, Alexander Sax, William Shen, Leonidas J Guibas, Jitendra Malik, and Silvio Savarese.
> Taskonomy: Disentangling task transfer learning. In CVPR, pp. 3712–3722, 2018.
>
> [4] François Chollet. Xception: Deep learning with depthwise separable convolutions. In Proceedings of
> the IEEE conference on computer vision and pattern recognition, pp. 1251–1258, 2017.
>
> [5] Johannes Ballé, David Minnen, Saurabh Singh, Sung Jin Hwang, and Nick Johnston. Variational image compression with a scale hyperprior. In ICLR, 2018.
>
> [6] Zhengxue Cheng, Heming Sun, Masaru Takeuchi, and Jiro Katto. Learned image compression with discretized gaussian mixture likelihoods and attention modules. In CVPR, pp. 7939–7948, 2020.
>
> [7] Renjie Zou, Chunfeng Song, and Zhaoxiang Zhang. The devil is in the details: Window-based
> attention for image compression. In CVPR, pp. 17492–17501, 2022.
>
> [8] Yinhao Zhu, Yang Yang, and Taco Cohen. Transformer-based transform coding. In ICLR, 2021.
>
> [9] Ming Lu, Peiyao Guo, Huiqing Shi, Chuntong Cao, and Zhan Ma. Transformer-based image
> compression. In DCC, pp. 469–469. IEEE, 2022.

---

> ### Author Response · Authors · 2024-12-01
> **Official Response to Reviewer 5H7f (Part 2/7)**
>
> **Q3: No results with time and space complexity.**
>
> **A3:** Our work uses multi-task learning algorithms for effective data encoding, bridging the gap between representation learning and compression. We recognize that reducing time and space complexity is a crucial consideration for real-world applications, even if our main focus is on the advantages of multi-task learning for compression. In  **A.4** [revised appendix, L.1115-1249], we provided complexity analysis of the Lookahead Module and Taskonomy-Aware Compression Module. We also reproduce the following:
>
> 1. **Lookahead Module Complexity**
>
>    The Lookahead module consists of two main components: task grouping and DAG construction.
>
>    1.1 **Task Grouping Complexity:**
>
>    Task grouping involves several steps:
>
>    - **Computing Pairwise Coherence:** This requires evaluating $N^2/2$ task pairs, which has a time complexity of $O(N^2)$.
>
>    - **Higher-Order Task Groupings Coherence:** After calculating the pairwise coherence, we compute the coherence of higher-order groups, which has exponential complexity represented as $O(N^N)$, where $N$ is the number of tasks. We estimate the coherence of higher-order group sets based on pairwise coherence scores. The computational overhead associated with this step is minimal and can be considered nearly negligible.
>
>    - **Optimization for Multi-Task Network Selection:** Finally, we select the optimal subset of $K$ multi-task networks from the available task groups. This step is computationally challenging, as it involves solving an NP-hard optimization problem under the given budget constraints. The complexity of this optimization problem is:
>      $$
>      O(\vert \mathcal{T} \vert \cdot \vert \boldsymbol{C}_0 \vert^{\frac{K}{Q}})
>      $$
> 		where $\vert \mathcal{T} \vert$ is the number of tasks, $\vert \boldsymbol{C}_0 \vert$ is the number of candidate networks, $K$ is the budget, and $Q$  is the minimum cost of a network in $C_0$. To solve this NP-hard problem efficiently, we employ a branch-and-bound approach. The prototype implementation of the algorithm can be found in [12].
>
>         [12] https://github.com/tstandley/taskgrouping/blob/master/network_selection/main.cpp
>
>    1.2 **DAG Construction:** After task grouping, the Lookahead module constructs a DAG based on conditional entropy. The complexity of calculating conditional and independent entropies in this stage is $O(K^2 \cdot n \cdot m)$, where $K$ is the number of task groups, $n$ is the complexity of calculating conditional entropy, $m$ is the complexity of calculating independent entropy for each group.
>
> 	1.3. **Overall Complexity of Lookahead Module:** Although complexity increases with the number of tasks ($N$) and task groups ($K$), inspired by traditional encoding lookahead modules, using low-resolution inputs and a lightweight feature extraction backbone during pre-analysis mitigates this.
>
> 2. **Taskonomy-Aware Compression Module Complexity**
>
> 	| | **Task Encoder $l_a$**   | **Task Decoder $l_s$**     |
> 	|-------------------------------------------|----------|------------|
> 	| **KParams**                              |  16467.2  |  525.1    |
> 	| **MMACs**                                 |  25708.0  |  4968.1   |
>
> 	*Table 1: Parameters and Forward MACs of Xception-based  Encoder/Decoder of Task on 512 × 512 images.*
>
> 	| **Context Module**           | **$\boldsymbol{\theta}_{hd}$**    | **$\boldsymbol{\theta}_{cm}$**    | **$\boldsymbol{\theta}_{cm+}$**   | **$\boldsymbol{\theta}_{gc}$**    |
> 	|------------------------------|------------|------------|------------|------------|
> 	| **KParams**                  |  5810.1   |  755.2    |  1487.7   |  2264.4   |
> 	| **MMACs**                     |  8925.1   |  1148.2  |  2264.4   |  7100.5   |
>
> 	*Table 2: Parameters and Forward MACs of entropy context modules on 512 × 512 images.*
>
> 	In Taskonomy-Aware Compression Module, the process is divided into two stages: task analysis and feature compression. The feature extraction and compression  grows linearly with the number of task groups $K$. The feature decoder phase grows linearly with the number of tasks ($N$). The total complexity is:$$\text{Total Complexity of Compression} = N \cdot l_s + K \cdot (l_a+\theta_{hd}+\theta_{cm+}+\theta_{gc})$$Substituting in the specific parameters:$$\text{Total Complexity of Compression} = N \cdot 4968.1 + K \cdot 40088.0$$
>
> 3.  **Summary of Complexity**
>
>     | Module                         | Complexity                             | Dominant Term |
>     |---------------------------------|----------------------------------------|---------------|
>     | **Lookahead Module**            | $O(N^2) + O(K^2 \cdot n \cdot m)$    | $O(N^2)$    |
>     | **Taskonomy-Aware Compression  Mode**                    | $O(N \cdot l_s + K \cdot (l_a+\theta_{hd}+\theta_{cm+}+\theta_{gc}))$ | $O(N)$ |

---

> ### Author Response · Authors · 2024-12-01
> **Official Response to Reviewer 5H7f (Part 3/7)**
>
> **Q4. Lack of baselines for comparison, I believe that additional  comparisons are necessary to clarify the consistent robustness of the proposed method compared to other approaches.**
>
> Thank you for your valuable feedback. we have included additional comparisons and ablation experiments in response to your feedback and the recommendations of Reviewer 5H7f:
>
> **1. Ablation 1: An end-to-end compression with multiple supervised tasks as auxiliary tasks.**
>
>    In  **A.3.1** [revised appendix,  L.880-956], we’ve added an ablation study to  integrate multiple supervised tasks as auxiliary losses into the end-to-end compression framework.
>
>    1.1 **Experiment Setting**: We trained 6 tasks (Semantic Segmentation, Surface Normal, Edge Texture, Depth Z-buffer, Keypoint 2D, and Autoencoder) using a combined loss function that includes both compression and task losses:$$L =L_{compression} + \sum_i w_i \cdot L_{task_i}.$$ Due to limited time and computational resources for fine-tuning the weights, we set equal weights $w_i=1$ for each task.
>
>    1.2  **Key Results**: We compared our method with baseline methods (MLIC++ and Task Auxiliary+MLIC++).  The Performance-Rate curves are shown in **Fig.9** [revised appendix L.883-899].  We also present the key results in the table below.  Our method consistently outperforms these baselines, which emphasizes the value of task grouping and causal DAG.
>
> | **Method**      | **Semantic Seg.** |  **Depth Z-buffer** |  **Surface Normal** |  **Keypoint 2D** |  **Edge Texture** |  **Autoencoder**  |
> |-----------------|-------------------|--------------------|--------------------|-----------------|-----------------|----------------|
> | **MLIC++**     | -24.72% | 11.79% | 8.52% | 32.99% | 3.59% | -36.32% |
> | **Task Auxiliary+MLIC++**     | -33.35% | -8.00% | -27.98%| 30.44% | -28.08% | **-42.63%**  |
> | **Ours**     | **-98.10%** | INF | **-99.37%** | **-91.38%** | **-88.47%** | 1.49% |
>
> *Table 1. BD-Performance. VTM-17.0 is used as the anchor. 'INF' indicates that the Performance-Rate curves do not intersect, thus no valid results can be computed.*
>
>
>    1.3  **Insights**: As shown in **Fig. 9** [revised main paper L.883-899], using auxiliary tasks (Task Auxiliary+MLIC++) gives useful insights into task interactions.
>
>    - Collaboration: For tasks like **Edge Texture**, **Semantic Segmentation**, and **Surface Normal**, the auxiliary task approach helps improve performance by encouraging better task collaboration.
>
>    - Unique: For **Autoencoder**, the performance is similar to the end-to-end method.
>
>    - Conflict: *However, we also observed that* **Keypoint 2D** *sees performance degradation with the auxiliary task approach, which shows why task grouping is essential to avoid conflicts.*
>
> **2. Ablation 2: Single task groups instead of grouping tasks together in phase 1.**
>
>    We added ablation study in **A 3.2** [revised appendix, L.958-1021].
>
>    2.1 **Experiment Setting**: We added an ablation study comparing single-task models with two task grouping models: **Group 1** (Semantic Segmentation, Depth Z-buffer, Surface Normal) and **Group 2** (Surface Normal, Keypoint 2D).  The BD-Performance comparison is shown in **Table 2** [revised appendix, A 3.2, L.972-986] and further visualized in **Fig. 10** [revised appendix, A 3.2, L.988-1008].
>
>    2.2 **Key Findings:**
>
>    - **Group 1** :  **Semantic Segmentation** and **Depth Z-buffer** performs better compared to the single-task models.
> 	- **Group 2** :  **Keypoint 2D** shows significant improvements compared to the single-task models.
> 	-  **Surface Normal**: the performance of grouped models is slightly inferior to single-task models.
>
>    2.3 **Insights:**
>
>    - Task grouping reduces redundancy and improves performance by sharing encoders and representations across tasks.
>
>    - An additional benefit is that the shared encoder only needs to perform inference once. For example, in **Group 1**, a single encoder is used to extract features for all 3 tasks, resulting in a total bitrate of **0.0015 bpp**. In contrast, treating each task independently requires 3 separate feature extractions, leading to a total bitrate of **0.0028 bpp**.
>
>    This study reinforces the benefits of task grouping for better compression efficiency and task collaboration. We hope this clarifies your concern and strengthens our approach.

---

> ### Author Response · Authors · 2024-12-01
> **Official Response to Reviewer 5H7f (Part 4/7)**
>
> **3. Ablation 3: Explore the impact when the same task is not allowed to appear in different clusters.**
>
>    In **A.3.3** [revised appendix, L.1022-1041], we conducted experiments under two setups: In Setting 1, tasks can't appear in different clusters, and in Setting 2 they can. As shown in **Table 3** [revised appendix, L.1009-1016] and **Table 4** [revised appendix, L.1026-1035], the Setting 2 achieves better performance, highlighting the benefits of task interdependence. While there’s no extra inference complexity (since each task is still only processed once during test), it does lead to higher GPU memory usage during training. So, it's a trade-off between performance gains and resource consumption.
>
> **4. Ablation 4: Comparasion to generative baselines such as VQGAN.**
>
>    In **A.3.5** [revised appendix, L.1068-1114] and **Fig.12** [revised appendix, L.1080-1102], we provide an ablation study comparing our method to VQGAN, which highlights the fundamental differences in both optimization goals and performance.
>
>    4.1 **Distinction in  Optimization Goals:** VQGAN-based compression methods [10][11] focus on optimizing perceptual compression. In contrast, our method diverges significantly by targeting efficient multi-task semantic compression. We specifically optimize for the shared use of compact semantic representations across multiple tasks, which reduces redundancy and enhances the compression efficiency.
>
>    4.2 **Distinction in  Performance:**
>    As illustrated in **Fig.12** [revised appendix, L.1080-1102], VQGAN introduces reconstruction errors due to its generative approach. These errors result in discrepancies from the ground truth, limiting its performance in tasks requiring precise feature preservation. For example, tasks like **Keypoint 2D detection**  require capturing fine-scale, sparse local features. These tasks are challenging for VQGAN, as its generative nature is less suited to applications that require high precision.
> In contrast, our method not only achieves superior compression efficiency at low bitrates but also maintains near-optimal performance at higher bitrates, closely aligning with the supervision anchor.
>
>    In conclusion, the key advantage of our approach lies in its ability to compress semantic information across tasks required for high precision. VQGAN is based on a generative framework and focuses on perceptual quality results in inferior fidelity performance.
>
>
>    [10] Patrick Esser, Robin Rombach, and Bjorn Ommer. Taming transformers for high-resolution image
>    synthesis. In Proceedings of the IEEE/CVF conference on computer vision and pattern recognition,
>    pp. 12873–12883, 2021.
>
>    [11] Qi Mao, Tinghan Yang, Yinuo Zhang, Zijian Wang, Meng Wang, Shiqi Wang, Libiao Jin, and Siwei
>    Ma. Extreme image compression using fine-tuned vqgans. In 2024 Data Compression Conference
>    (DCC), pp. 203–212. IEEE, 2024.

---

> ### Author Response · Authors · 2024-12-01
> **Official Response to Reviewer 5H7f (Part 5/7)**
>
> **5. Additional comparison with MLIC++.**
>
> In the task of compression for multiple downstream tasks, we investigate two compression paradigms:
> **Analysis and Then Compression (ATC)** and **Compression and Then Analysis (CTA)**.
>
> Our approach follows the **ATC** paradigm, which consists of two main stages: (1) the **downstream task analysis phase**, where the input image is processed for tasks such as keypoint detection, and (2) the **feature compression phase**.
>
> In contrast, MLIC++ adopts the **CTA** paradigm, where the process is reversed: (1) the **image compression phase**, and (2) the **downstream task analysis phase**.
>
> To better illustrate the computational costs involved in each approach, we summarize the parameters and forward MACs for the different components in the ATC and CTA pipelines, presented in Tables 1, 2, and 3 below.
>
> ---
>
> | **MLIC++ Encoder/Decoder Module** | **$g_a$**   | **$g_s$**     |
> |-----------------------------------|----------|------------|
> | **KParams**                       |  12033.6  |  4396.3    |
> | **MMACs**                          |  194556.9  |  296377.9 |
>
> *Table 1: Parameters and Forward MACs of Encoder/Decoder of MLIC++ on 512 × 512 images.*
>
> ---
>
> | **Task Encoder/Decoder Module** | **$l_a$**   | **$l_s$**     |
> |-------------------------------------------|----------|------------|
> | **KParams**                              |  16467.2  |  525.1    |
> | **MMACs**                                 |  25708.0  |  4968.1   |
>
> *Table 2: Parameters and Forward MACs of Task Encoder/Decoder of Xception on 512 × 512 images.*
>
> ---
>
> | **Context Module**           | **$\boldsymbol{\theta}_{hd}$**    | **$\boldsymbol{\theta}_{cm}$**    | **$\boldsymbol{\theta}_{cm+}$**   | **$\boldsymbol{\theta}_{cg}$**    |
> |------------------------------|------------|------------|------------|------------|
> | **KParams**                  |  5810.1   |  755.2    |  1487.7  |  2264.4   |
> | **MMACs**                     |  8925.1   |  1148.2   |  2264.4   |  7100.5   |
>
> *Table 3: Parameters and Forward MACs of entropy context modules on 512 × 512 images.*
> | **Method**      |   **Keypoint 2D** |  **Edge Texture** |
> |-----------------|-----------------|-----------------|
> | **Ours wo $\boldsymbol{\theta}_{cm+}$**     |  -65.79% | -66.35% |
> | **Ours wo $\boldsymbol{\theta}_{cg}$**     | -38.37% | -38.04% |
> | **Ours**     | **-91.38%** | **-88.47%** |
>
> *Table 4. BD-Performance. VTM-17.0 is used as the anchor. 'INF' indicates that the Performance-Rate curves do not intersect, thus no valid results can be computed.*
>
>    5.1 **Analysis and Then Compression (ATC) Mode Complexity**
>
>    In **ATC mode**, the process is divided into two stages: task analysis and feature compression. The task analysis phase involves task encoder which grows linearly with the number of task groups $K$, and task decoder which grows linearly with the number of tasks $N$. The feature compression grows linearly with the number of task groups $K$. The total complexity of the ATC mode is:
>
>    $$
>    \text{Total Complexity of ATC} = N \cdot l_s + K \cdot (l_a + \theta_{hd} +\theta_{cm+} + \theta_{gc})
>    $$
>
>    Substituting in the specific parameters:
>
>    $$
>    \text{Total Complexity of ATC} = N \cdot 4968.1 + K \cdot 40088.0
>    $$
>
>    Thus, the computational cost of the ATC mode grows linearly with $N$, while the task grouping and context module overhead grows with $K$, the number of task groups.
>
>    5.2 **CTA (Compression Then Analysis) Mode Complexity**
>
>    In **CTA mode**, the total complexity comes from both the compression and task analysis stages. The compression stage's complexity is dictated by the encoder/decoder modules, while the task analysis phase grows linearly with $N$. The total complexity is given by:
>
>    $$
>    \text{Total Complexity of CTA} = g_a + g_s + \theta_{hd} + \theta_{cm} + \theta_{gc} + N \cdot (l_a + l_s)
>    $$
>
>    Substituting in the parameter values:
>
>    $$
>    \text{Total Complexity of CTA} = 500008.6 + N \cdot 30676.1
>    $$
>
>    5.3 **Summary of Complexity**
>
>    Compared to ATC, CTA has a fixed overhead from the compression stage, and its complexity also grows linearly with $N$, but with a higher constant cost due to the compression step.
>
>    | Module                         | Complexity                             | Dominant Term |
>    |---------------------------------|----------------------------------------|---------------|
>    | **Lookahead Module**            | $O(N^2) + O(K^2 \cdot n \cdot m)$    | $O(N^2)$    |
>    | **ATC Mode**                    | $O(N \cdot l_s + K \cdot (l_a + \theta_{hd} + \theta_{cm+} + \theta_{gc}))$ | $O(N)$ |
>    | **CTA Mode**                    | $O(g_a + g_s + \theta_{hd} + \theta_{cm} + theta_{gc} + N \cdot (l_a + l_s))$ | $O(N)$ |

---

> ### Author Response · Authors · 2024-12-01
> **Official Response to Reviewer 5H7f (Part 6/7)**
>
> **Q5: Could the task grouping process be further explained?**
>
> **A5:** Task grouping involves 3 steps:
>
>    **Step1: Computing Pairwise Coherence:** First, we compute the coherence scores for each pair of tasks.
>    For a given training batch $\mathcal{X}^t$ at time-step $t$, **$\Theta_{g \vert u}^{t+1}$** in **Eq. 7** [revised main paper, L.256] represents the updated shared parameters after a gradient step with respect to task $i$:
>
>    $$
>    \Theta_{g \vert u}^{t+1} = \Theta_{g}^{t} - \eta \nabla_{\Theta_g^t} L_u(\tau_u|\mathcal{X}^t, \Theta_g^t, \Theta_u^t)\, .
>    $$
>
>    Next, in **Eq. 8** [revised main paper, L.262], we calculate the **forecast loss** for each task by using the updated shared parameters, while keeping the task-specific parameters and input batch fixed. This allows us to assess the effect of the gradient update from task $u$ on task $v$.
>    We compare the loss of task $v$ before and after applying the gradient update from task $u$ to the shared parameters:
>    $$
>    C^t_{u \to v} = 1 - \frac{L_{v}(\tau_v|\mathcal{X}^t, \Theta_{g \vert u}^{t+1}, \Theta_v^t)}{L_{v}(\tau_v|\mathcal{X}^t, \Theta_g^t, \Theta_v^t)}\, .
>    $$
>    A positive value of $C^t_{u \to v}$ indicates that the update to the shared parameters results in a lower loss for task $v$, while a negative value suggests that the shared parameter update is detrimental to the performance of task $v$.
>
>    This requires evaluating $N^2/2$ task pairs, which has a time complexity of $O(N^2)$. The quadratic nature of this computation means that as the number of tasks $N$ increases, the cost grows rapidly.
>
>
>    **Step2: Higher-Order Task Groupings Coherence:** After calculating the pairwise coherence, we compute the coherence of higher-order groups, which has exponential complexity represented as $O(N^N)$, where $N$ is the number of tasks. We estimate the coherence of higher-order group sets based on pairwise coherence scores. This strategy called Higher Order Approximation (HOA) is introduced in Section 5.3.2 of preliminary work [2]. This approach approximates higher-order task groupings based on pairwise task performance, and reduces computation complexity by 45%.  In Section 6 of [2] empirically demonstrated that HOA can effectively approximate the optimal solution, even though it lacks formal theoretical derivation.
>
>    In the revised version **A.3.4** [revised appendix, L. 1042-1068], we provide further validation. As shown in **Fig. 11** [revised appendix, L. 1045-1062], we observe that although the coherence score between tasks is not strictly symmetric ($ C_{u \to v} \neq  C_{v \to u}$), it tends to exhibit strong symmetry in practice, allowing us to approximate values while maintaining accuracy.
>
>
>    The computational overhead associated with this step is minimal and can be considered nearly negligible.
>
>    **Step3: Optimization for Multi-Task Network Selection:** Finally, we select the optimal subset of $K$ multi-task networks from the available task groups. This step is computationally challenging, as it involves solving an NP-hard optimization problem under the given budget constraints. The complexity of this optimization problem is:
>      $$
>      O(\vert \mathcal{T} \vert \cdot \vert \boldsymbol{C}_0 \vert^{\frac{K}{\min_{n \in \boldsymbol{C}_0} c_n}})
>      $$
> 		where $\vert \mathcal{T} \vert$ is the number of tasks, $\vert \boldsymbol{C}_0 \vert$ is the number of candidate networks, $K$ is the budget, and $\min_{n \in \boldsymbol{C}_0} c_n$ is the minimum cost of a network in $C_0$.
>
>    In the revised version **A.3.5** [revised appendix, L.1249-1274], we provide implementation details for task grouping. We adopt a branch-and-bound method, similar to prior works [2], to find the optimal solution (as shown in **Algorithm 2** [revised appendix, L.1188-1219]) of this NP-hard problem. The prototype implementation of the algorithm can be found in [12].
>
> ---
>
> [2] Trevor Standley, Amir Zamir, Dawn Chen, Leonidas Guibas, Jitendra Malik, and Silvio Savarese. Which tasks should be learned together in multi-task learning? In International conference on machine learning, pp. 9120–9132. PMLR, 2020.
>
> [12] https://github.com/tstandley/taskgrouping/blob/master/network_selection/main.cpp

---

> ### Author Response · Authors · 2024-12-01
> **Official Response to Reviewer 5H7f (Part 7/7)**
>
> **Q6: Clarity of Parent Node Selection**
>
> **A6:** Thank you for your valuable comment. You are correct that the explanation could be clearer. In our approach, **the parent node for a child node is selected by minimizing the conditional entropy of the child node, with an additional condition that the parent node must have a lower information entropy than the child node.** This ensures that the causal relationships are consistent and meaningful.
>
> We acknowledge that this detail may not have been sufficiently clear in the original description. In the revised version of **Algorithms 1** [revised main paper, L.290-309], we have clarified the conditions for parent node selection to provide a more accurate and transparent explanation of this process.
>
> ---
>
> **Q7: More explanation regarding the bit-rate for each task**
>
> **A7:**  Thank you for this question. To clarify, the bitrates for each task are reported in **Sec 4.3.1 Performance of Compression for Multiple Tasks** (Fig. 5 [revised main paper, L.393-417]). The bitrate-performance curve shows task bitrates on the x-axis.
>
>  The task grouping module yields the following groupings:
> - **Group 1**: (**Semantic Seg**., **Depth Z-buffer**, **Surface Normal**)
> - **Group 2**: (Surface Normal, **Keypoint 2D**)
> - **Group 3**: (Semantic Seg., Surface Normal, **Edge Texture**, **Autoencoder**)
>
> **Bolded** tasks are decoded during the validation/test phase, while the **skipped** tasks are only used during training to optimize the joint training of tasks within the same group.  Tasks in the same group share a common bitrate, as they leverage the same compressed representations. The causal graph structure is defined as: **Group 1 → Group 2 → Group 3**.
>
> To improve clarify, we reorganized **Fig. 5** [revised main paper, L.393-417] into the following table format, showing bitrate and test loss values for each task:
>
> | **Method**      | **Semantic Seg.** |  | **Depth Z-buffer** |  | **Surface Normal** |  | **Keypoint 2D** |  | **Edge Texture** |  | **Autoencoder** |  |
> |-----------------|-------------------|--|--------------------|--|--------------------|---|-----------------|--|-----------------|--|----------------|--|
> |                 | **Bitrate** | **Test Loss** | **Bitrate** | **Test Loss** | **Bitrate** | **Test Loss** | **Bitrate** | **Test Loss** | **Bitrate** | **Test Loss** | **Bitrate** | **Test Loss** |
> | **Group 1**     | 0.0006 | 0.1819 | 0.0006 | 0.5906 | 0.0006 | 0.2372 | - | - | - | - | - | - |
> |                 | 0.0055 | 0.1777 | 0.0055 | 0.5800 | 0.0055 | 0.1857 | - | - | - | - | - | - |
> |                 | 0.0117 | 0.1768 | 0.0117 | 0.5695 | 0.0117 | 0.1762 | - | - | - | - | - | - |
> | **Group 2**     | - | - | - | - | Skipped | Skipped | 0.0047 | 0.2523 | - | - | - | - |
> |                 | - | - | - | - | Skipped | Skipped | 0.0394 | 0.2512 | - | - | - | - |
> |                 | - | - | - | - | Skipped | Skipped | 0.0671 | 0.2512 | - | - | - | - |
> | **Group 3**     | Skipped | Skipped | - | - | Skipped | Skipped | - | - | 0.0133 | 0.2855 | 0.0133 | 0.0333 |
> |                 | Skipped | Skipped | - | - | Skipped | Skipped | - | - | 0.1449 | 0.2799 | 0.1449 | 0.0149 |
> |                 | Skipped | Skipped | - | - | Skipped | Skipped | - | - | 0.2495 | 0.2792 | 0.2495 | 0.0131 |
>
> From this table, we can observe the bitrate ranges for each task:
> - **Semantic Seg., Depth Z-buffer, Surface Normal**: 0.0006–0.0117 bpp
> - **Keypoint 2D**: 0.0047–0.0671 bpp
> - **Edge Texture, Autoencoder**: 0.0133–0.2495 bpp
>
> **This demonstrates that simpler tasks (e.g., Semantic Seg., Depth Z-buffer) are allocated a subset of the latent space (base layer), while more complex tasks (e.g., Keypoint 2D, Edge Texture) utilize additional subsets (enhancement layers). This scalable encoding aligns with our task grouping and conditional entropy causal graph framework.**
>
>  **Training Process**
>
> During training, bitrates are controlled via the rate-distortion trade-off, as described in**Eq. 4** (revised main page, L.156).  The trade-off is expressed as:$$\lambda \times \mathcal{D} + \mathcal{R},$$ where $\mathcal{D}$ represents distortion and  $\mathcal{R}$ denotes the bitrate. The weight $\lambda$ is empirically tuned to balance the trade-off, with typical values ranging from [0.04, 0.072, 0.14, 1, 1.932]. This ensures models are trained across a range of bitrates, from low-bitrate to near-lossless.
>
> Task-specific weights $w_i$ are adjusted based on the uncompressed performance to maintain consistent loss magnitudes across tasks, aiding joint training convergence. Therefore, the weight for each task at a given bitrate is determined by $\lambda \cdot w_i$.
>
> ---
>
> **Q8:  Appendix A.5.4 and A.5.5  not directlfocus more directly on the key concepts**
>
> **A8:**  Thank you for your advice. We have revised our manuscript and included only the necessary formula proofs that are critical to the core logic and flow of the paper.

---

> > ### Comment · Reviewer_5H7f · 2024-12-02
> >
> > Dear authors,
> >
> > I warmly thank the authors for addressing many of my concerns. However, after reading the revision, additional concerns have arisen, and I have decided to maintain my initial rating for now. I will give further consideration to this paper and finalize my rating after discussions with the other reviewers.
> >
> > The revised version of the paper provides much more information compared to the previous version and includes detailed explanations of the implementation. However, the following aspects appear to require additional revision:
> > - In the framework presented in Figure 2, the modules representing the proposed methodology are not sufficiently highlighted. Providing clearer information on the method used to link parent and child tasks would highlight the novelty of the proposed method.
> > - To enhance the overall quality of the paper, it would be beneficial to refine unnecessary citations (particularly LN 184-202) and incorporate the results currently placed in the appendix into the main body of the text.
> > - (minor) Some of the expressions in Algorithm 1, such as `min_cond_entropy`, could be revised for improved clarity.

---

> ### Author Response · Authors · 2024-12-03
>
> Dear Reviewer 5H7f,
>
> We are pleased to have addressed many of your concerns in the revised manuscript. We sincerely appreciate your detailed suggestions and will carefully address each of your additional concerns below:
>
> 1. **Highlighting the proposed method in Figure 2**:
>    Figure 2, which has been newly added in the revision, provides an overview of our extension from E2E Compression to Taskonomy-Aware Compression, clarifying the transition in Section 3.3 (*Causal Graph-Based Compression Method*, [L.280-357]).
>    In future revisions of Figure 2, we will highlight the process of selecting parent nodes that minimize the child node’s conditional entropy, with the additional constraint that the parent’s information entropy is lower than the child’s.
>
> 2. **Refining unnecessary citations and incorporating results from the appendix into the main body**:
>    We will carefully review the citations and retain only the most relevant references to avoid redundancy. Additionally, we will move the critical results currently in the appendix to the main body of the text to enhance the robustness of the experiments, demonstrate the effectiveness of the proposed method, and highlight the logical coherence of our approach.
>
> 3. **Improving clarity in Algorithm 1**:
>    We will revise the expressions to use more precise notation, *e.g.*, min_cond_entropy-> $\min_{Y_p \in \mathcal{Y}, H(Y_c) > H(Y_p)} \ H(Y_c \mid Y_p)$, in future revisions to improve clarity.

---

### Official Review · Reviewer_mF5B · 2024-11-03

**Soundness:** 3
**Presentation:** 1
**Contribution:** 3
**Rating:** 5
**Confidence:** 4

**Summary:**

This paper proposes a framework, Taskonomy-Aware Multiple Task Compression (TAMC), for lossy compression where the distortion is defined by multiple downstream tasks. First, TAMC  groups tasks into clusters where tasks within a cluster are mutually supportive and a share representation is learned for each cluster. Then, it leverages causal discovery to identify dependencies between groups. This results in a directed acyclic graph (DAG), which can be used for further compression of the representation.  Experiments on the Taskonomy dataset demonstrate that TAMC achieves superior bitrate reduction and task performance compared to baseline compression methods.

**Strengths:**

The paper discusses an important but less studied aspect of representation learning - how to leverage different supervised signals to learn a better representation, where "better" is defined as lower bitrate with higher downstream performance. This principle is sound. The experimental results demonstrate that the proposed framework achieves better performance comparing to end-to-end compression methods using the same bitrate.

**Weaknesses:**

**Clarity**: the description of the framework lacks clarity. Given that the paper proposes a fairly complex system, clarity is even more important for the readers to understand. For example, although there is a space constraint, the description of step 3 (taskonomy-based compression) is too brief. Other more specific questions are raised below.

**Lack of ablations**: there are at least 2 ablations the authors could provide. 1. an end-to-end compression with multiple supervised tasks as auxiliary tasks; 2. single task groups, where instead of grouping tasks together in phase 1, simply treat each task as a group and carry out the rest of the learning.

**Questions:**

1. Could you state the number of parameter sets used and explain how they are allocated across groups and individual tasks? The paper seems to suggest it's per group, but it also needs a set of task parameters per task for computing the pairwise gradient coherence.
2. How is the forecast loss in Eq.3 used?
3. Line 287, could you clarify if task order impacts the cost calculation and, if so, how do you address this potential variability in their method?
4. Is there any constraint on the subsets used for set cover?
4. Is it better or worse when the same task is not allowed to appear in different clusters?
5. How is bitrate controlled in the proposed framework? How do you determine $\lambda_i$ for each task?
6. Is the causal discovery step required? Could the minimum description length principle be used for causal discovery and, thus, unifying step 2 and 3 as finding a DAG structure that minimizes the bit-rate?
7. Could you provide a complexity analysis showing how computational requirements change as the number of tasks increases?
8. A simple baseline would be multi-task learning naively combined with end-to-end compression, i.e. without the grouping, just use them auxiliary tasks with well-tuned weights. Why would you not include such a baseline? How would this baseline compare theoretically to the proposed method?
9. Figure 3, could you add a legend for what the color shade represent?  Could you define the "Anchor" baseline in the figure caption or main text?
10. How does the method perform compare to more familiar baselines such as VQGAN?
11. Github link is not provided?
12. L137 typo "Hu"?

---

> ### Author Response · Authors · 2024-12-01
> **Official Response to Reviewer mF5B  (Part 1/7)**
>
> Dear Reviewer mF5B,
>
> Thank you for your thoughtful feedback. We really appreciate your constructive comments, which will help improve the clarity and completeness of the paper. In the revised version, we've added more details and experiments to address your concerns.
>
> In summary, key experiments and their corresponding QA are listed as:
> 1.  Ablation 1, QA 2: An end-to-end compression with multiple supervised tasks as auxiliary tasks, in  **A.3.1** [revised appendix, L.880-956].
> 2.  Ablation 2, QA 4: Single task groups instead of grouping tasks together, in  **A.3.2** [revised appendix,  L.880-956].
> 3.  Parameter sets, QA 5.
> 4.  Task order impacts, QA 7, in **A.3.4** [revised appendix, L.1068-1114].
> 5.  The same task is not allowed to appear in different clusters, QA 8, in **A.3.3** [revised appendix, L.1022-1041].
> 6. Complexity analysis, QA 12, in  **A.4** [revised appendix, L.1115-1249].
> 7. Comparison to VQGAN, QA 14,  in **A.3.5** [revised appendix, L.1068-1114]
>
>
> ---
>
>
> **Q1: The description of step 3 lacks clarity.**
>
> **A1:** Thank you for pointing that out. We agree that the description of Step 3 (Taskonomy-Aware Compression) could benefit from more detail. Our revisions include the following:
>
> 1.  In the **Related Work** Section [revised main paper, L.132-161], we have added background on end-to-end compression, with **Eq.1-4** formally defining the compression problem. This provides the foundation for a detailed explanation of Taskonomy-Aware Compression in Section 3.3 [revised main paper, L.280-357].
> 2.  **Fig. 2** [revised main paper, L.108-117] offers an overview of how we extend end-to-end compression to Taskonomy-Aware Compression, making the transition clearer.
> 3.  The Section **Scalable Compression Using the Causal DAG** [revised main paper, L.312-357] provides detailed implementation aspects of Taskonomy-Aware Compression using the Causal DAG. The cross-task local causal context is illustrated in **Fig. 4(a)** and formalized in **Eq. 11**, with differences from **Eq.3** marked in blue. The global causal context is shown in **Fig. 4(b)** and formalized in **Eq. 12-13**, with differences from **Eq.3** marked in green.
>
> We hope this added detail strengthens the clarity of step 3.
>
> ---
>
> **Q2: Lack of ablation 1: An end-to-end compression with multiple supervised tasks as auxiliary tasks.**
>
> **A2:**  In  **A.3.1** [revised appendix, L.880-956], we’ve added an ablation study to integrate multiple supervised tasks as auxiliary losses into the end-to-end compression framework.
> 1. **Experiment Setting**: We trained 6 tasks (Semantic Segmentation, Surface Normal, Edge Texture, Depth Z-buffer, Keypoint 2D, and Autoencoder) using a combined loss function that includes both compression and task losses:$$L =L_{compression} + \sum_i w_i \cdot L_{task_i}.$$ Due to limited time and computational resources for fine-tuning the weights, we set equal weights $w_i=1$ for each task.
>
> 2. **Key Results**: We compared our method with baseline methods (MLIC++ and Task Auxiliary+MLIC++).  The Performance-Rate curves are shown in **Fig.9** [revised appendix L.883-899].  We also present the key results in the table below.  Our method consistently outperforms these baselines, which emphasizes the value of task grouping and causal DAG.
>
>     | **Method**      | **Semantic Seg.** |  **Depth Z-buffer** |  **Surface Normal** |  **Keypoint 2D** |  **Edge Texture** |  **Autoencoder**  |
>     |-----------------|-------------------|--------------------|--------------------|-----------------|-----------------|----------------|
>     | **MLIC++**     | -24.72% | 11.79% | 8.52% | 32.99% | 3.59% | -36.32% |
>     | **Task Auxiliary+MLIC++**     | -33.35% | -8.00% | -27.98%| 30.44% | -28.08% | **-42.63%**  |
>     | **Ours**     | **-98.10%** | INF | **-99.37%** | **-91.38%** | **-88.47%** | 1.49% |
>
>     *Table 1. BD-Performance. VTM-17.0 is used as the anchor. 'INF' indicates that the Performance-Rate curves do not intersect, thus no valid results can be computed.*
>
> 3. **Insights**: As shown in **Fig. 9** [revised main paper L.883-899], using auxiliary tasks (Task Auxiliary+MLIC++) gives useful insights into task interactions.
>
>     - Collaboration: For tasks like **Edge Texture**, **Semantic Segmentation**, and **Surface Normal**, the auxiliary task approach helps improve performance by encouraging better task collaboration.
>
>     - Unique: For **Autoencoder**, the performance is similar to the end-to-end method.
>
>     - Conflict: *However, we also observed that* **Keypoint 2D** *sees performance degradation with the auxiliary task approach, which shows why task grouping is essential to avoid conflicts.*
>
> We hope this additional experiment helps clarify the benefits of our task grouping approach and answers your concern.

---

> ### Author Response · Authors · 2024-12-01
> **Official Response to Reviewer mF5B  (Part 2/7)**
>
> ### **Q3: How would this baseline compare theoretically to the proposed method?**
>
> **A3:** Here is the theoretical comparison between the baseline and proposed method.
>
> ### 1. **Information Theory Background**
>
> To comprehend redundancy and compression efficiency, we rely on **Conditional Entropy**, **Mutual Information** and **Data Processing Inequality (DPI)**.
>
> - **Mutual Information** $I(X; Y)$: Measures shared information between $X$ and $Y$. High mutual information indicates redundancy, reducing compression efficiency.
>
> - **Conditional Entropy** $H(Y | X)$: Represents uncertainty in $Y$ given $X$. It reflects how effectively information is transferred between tasks, reducing redundancy for efficient information flow.
>
> - **DPI**: The principle of DPI asserts that in a Markov chain, the flow of information is constrained to a decrease, never an increase.
>
> ### 2. **Baseline Method Analysis**
>
> In the baseline method, all tasks share the same latent space $ Z $, leading to two issues:
>
> #### 2.1. **Task Collaboration, Conflicts, and Suboptimal Performance (Representation Learning Perspective)**
>
> The shared representation  $Z$ is optimized for reconstruction as well as multiple tasks, but maximizing mutual information $I(Y_i; Y_j | Z)$ between tasks does not guarantee optimal performance, as  $Z$ is not tailored for task-specific dependencies. This leads to inefficiencies in handling task collaboration and conflicts.
>
> #### 2.2. **Redundant Information Transmission (Mutual Information and DPI Perspective)**
>
> Tasks receive redundant information from $Z$. Since all tasks depend on $Z$, each task receives irrelevant information, causing **redundant transmission** and inefficiency in compression.
>
> The mutual information $I(Z; Y_i)$ for each task is unnecessarily high:
>
> $$
> I(Z; Y_i) \quad \forall i \in [1, N]
> $$
>
> This redundancy increases the conditional entropy $H(Y_i | Z)$, as $Y_i$ cannot fully rely on its unique features but is influenced by $Z$.
>
> According to DPI, redundant transmission increases entropy:
>
> $$
> H(Z) \geq H(Y_i | Z) \quad \text{for each task, leading to suboptimal compression.}
> $$
>
> Thus, since $Z$ is shared, the conditional entropy of $Y_i$ given $Z$ cannot be minimized, leading to inefficiency.
>
> ### 3. **Proposed Method Analysis**
>
> The proposed method consists of **task grouping** and **causal graphs**.  The two modules elucidate the disentangle, grouping and hierarchical structure of semantic representation, followed by directly decoding for multiple tasks. The analyze-then-compression paradigm effectively reduces redundancy according to DPI theory.
>
> #### 3.1. **Maximizing Mutual Information Within Task Groups**
>
> We maximize mutual information within each task group $G_k =$ {$ Y_1, Y_2, \dots, Y_k$}:
>
> $$
> \max_{\{ G_k \}} \sum_{k} \sum_{Y_i,Y_j \in G_k} I(Y_i; Y_j)
> $$
>
> This maximizes collaboration within each group, improving compression efficiency.
>
> #### 3.2. **Minimizing Conditional Entropy of Child Nodes Between Task Groups**
>
> We minimize the conditional entropy of child nodes between task groups. In the causal graph, each parent node $G_{\text{parent}(k)}$ provides context for child node $G_k$, reducing redundancy:
>
> $$
> \min \sum_{k} H(G_k | G_{\text{parent}(k)})
> $$
>
> #### 3.3. **Final Optimization Objective**
>
> The optimization objective is:
>
> $$
> \max_{\{ G_k \}} \sum_{k} \sum_{Y_i,Y_j \in G_k} I(Y_i; Y_j) - \lambda \sum_{k} H(G_k | G_{\text{parent}(k)})
> $$
>
> ### 4. **Conclusion**
> - **Multiple Task Representation**: In the baseline method, tasks are optimized independently using a shared latent space $Z$, leading to redundant representations and high mutual information $ I(Y_i; Y_j | Z)$, which limits task-specific learning and performance. The proposed method reduces redundancy by grouping tasks, minimizes irrelevant mutual information, and enhances task performance by sharing only relevant knowledge when mutually beneficial.
>
> - **Redundancy Modeling**: The baseline method uses a shared latent space $Z$ for all tasks, which leads to inefficiency as it does not allow for task-specific entropy minimization. This results in higher conditional entropy  $H(Y_i | Z)  $ for each task, causing unnecessary redundancy in encoding and transmission. The proposed method uses causal graphs to allow dependent groups to minimize conditional entropy and allow independent groups to focus on the unique features, lowering overall entropy and improving both compression efficiency and scalability.

---

> ### Author Response · Authors · 2024-12-01
> **Official Response to Reviewer mF5B (Part 3/7)**
>
> **Q4: Lack of ablation 2: Single task groups instead of grouping tasks together in phase 1.**
>
> **A4:**  We added ablation study in **A 3.2** [revised appendix, L.958-1021].
>
> **1. Experiment Setting**: We added an ablation study comparing single-task models with two task grouping models: **Group 1** (Semantic Segmentation, Depth Z-buffer, Surface Normal) and **Group 2** (Surface Normal, Keypoint 2D).  The BD-Performance comparison is shown in **Table 2** [revised appendix, A 3.2, L.972-986] and further visualized in **Fig. 10** [revised appendix, A 3.2, L.988-1008].
>
> **2. Key Results:**
>
> - **Group 1** :  **Semantic Segmentation** and **Depth Z-buffer** performs better compared to the single-task models.
> - **Group 2** :  **Keypoint 2D** shows significant improvements compared to the single-task models.
> -  **Surface Normal**: the performance of grouped models is slightly inferior to single-task models.
>
> **3. Insights:**
>
> - Task grouping reduces redundancy and improves performance by sharing encoders and representations across tasks.
> - An additional benefit is that the shared encoder only needs to perform inference once. For example, in **Group 1**, a single encoder is used to extract features for all 3 tasks, resulting in a total bitrate of **0.0015 bpp**. In contrast, treating each task independently requires 3 separate feature extractions, leading to a total bitrate of **0.0028 bpp**.
>
> This study reinforces the benefits of task grouping for better compression efficiency and task collaboration. We hope this clarifies your concern and strengthens our approach.
>
> ---
>
> **Q5: Could you state the number of parameter sets used and explain how they are allocated across groups and individual tasks?**
>
> **A5:** Our work explores the potential of achieving multi-task collaborative compression by disentangling semantic space representations, task grouping, and modeling a conditional entropy causal graph. To conduct our experiments, we made relatively ideal assumptions based on prior work in Multiple Task Learning (MTL) [1,2,3]. Specifically, we used Xception [7] as the encoder backbone for both single-task and grouped tasks. Each task has its own **task-specific decoder**, differing primarily in output channels (e.g., 3 channels for image reconstruction and 20 for semantic segmentation). The pretrained model can be accessed at [8].
>
> **1. Parameter Set Allocation**:
>
> To clarify the allocation of parameters across individual tasks and task groups, we present the following table:
>
> | **Xception Encoder/Decoder Module** | Encoder | Semantic Seg. Decoder | Depth Z-buffer Decoder | Surface Normal Decoder | Keypoint 2D Decoder | Edge Texture Decoder | Autoencoder Decoder |
> | --- | --- | --- | --- | --- | --- | --- | --- |
> | **KParams** | 16467.2 | 525.1 | 520.2 | 520.8 | 520.2 | 520.2 | 520.8 |
> | **MMACs** | 25708.0 | 4968.1 | 3684.6 | 3684.6 | 3684.6 | 3684.6 | 3684.6 |
>
> _Parameters and Forward MACs of Task Encoder/Decoder for Xception on 512 × 512 images._
>
> **2. Pairwise Gradient Coherence**:
>
> Each task has its own decoder parameters, while the encoder parameters are shared. During training, we compute pairwise gradient coherence between tasks to promote mutual information sharing in the bottleneck representations, while preserving task-specific decoders.
>
> In principle, different tasks have varying architectures based on computational needs and applications, we use a common encoder backbone and task-specific decoders for consistency across experiments.
>
> ---
>
> [1] Chris Fifty, Ehsan Amid, Zhe Zhao, Tianhe Yu, Rohan Anil, and Chelsea Finn. Efficiently identifying
> task groupings for multi-task learning. Advances in Neural Information Processing Systems, 34:
> 27503–27516, 2021.
>
> [2] Trevor Standley, Amir Zamir, Dawn Chen, Leonidas Guibas, Jitendra Malik, and Silvio Savarese.
> Which tasks should be learned together in multi-task learning? In International conference on
> machine learning, pp. 9120–9132. PMLR, 2020.
>
> [3] Amir R Zamir, Alexander Sax, William Shen, Leonidas J Guibas, Jitendra Malik, and Silvio Savarese.
> Taskonomy: Disentangling task transfer learning. In CVPR, pp. 3712–3722, 2018.
>
> [7] François Chollet. Xception: Deep learning with depthwise separable convolutions. In Proceedings of
> the IEEE conference on computer vision and pattern recognition, pp. 1251–1258, 2017.
>
> [8] https://drive.google.com/drive/folders/1XQVpv6Yyz5CRGNxetO0LTXuTvMS_w5R5

---

> ### Author Response · Authors · 2024-12-01
> **Official Response to Reviewer mF5B (Part 4/7)**
>
> **Q6: How is the forecast loss in Eq.3 [previous main page, L.273] used?**
>
> **A6:** Apologies for the confusion. Due to a citation error in our reference, the forecast loss is actually represented by **Eq. 4** in the original text, not **Eq. 3**. We have clarified this in the revised version. Specifically, **Eq. 3** in the previous version corresponds to **Eq. 7** [revised main paper, L.256], and **Eq. 4** in the previous version corresponds to **Eq. 8**[revised main paper, L.262].
>
> For a given training batch $\mathcal{X}^t$ at time-step $t$, **$\Theta_{g \vert u}^{t+1}$** in **Eq. 7** [revised main paper, L.256] represents the updated shared parameters after a gradient step with respect to task $i$:
>
> $$
> \Theta_{g \vert u}^{t+1} = \Theta_{g}^{t} - \eta \nabla_{\Theta_g^t} L_u(\tau_u|\mathcal{X}^t, \Theta_g^t, \Theta_u^t)\, .
> $$
>
> Next, in **Eq. 8** [revised main paper, L.262], we calculate the **forecast loss** for each task by using the updated shared parameters, while keeping the task-specific parameters and input batch fixed. This allows us to assess the effect of the gradient update from task $u$ on task $v$.
> We compare the loss of task $v$ before and after applying the gradient update from task $u$ to the shared parameters:
> $$
> C^t_{u \to v} = 1 - \frac{L_{v}(\tau_v|\mathcal{X}^t, \Theta_{g \vert u}^{t+1}, \Theta_v^t)}{L_{v}(\tau_v|\mathcal{X}^t, \Theta_g^t, \Theta_v^t)}\, .
> $$
> A positive value of $C^t_{u \to v}$ indicates that the update to the shared parameters results in a lower loss for task $v$, while a negative value suggests that the shared parameter update is detrimental to the performance of task $v$.
>
> ---
> **Q7: Line 287 [previous main page], could you clarify if task order impacts the cost calculation and, if so, how do you address this potential variability in their method?**
>
> **A7:** This is a great question! You're absolutely right to highlight the potential impact of task order, which could indeed influence the results. To clarify, the strategy called Higher Order Approximation (HOA) is introduced in Section 5.3.2 of preliminary work [2]. This approach approximates higher-order task groupings based on pairwise task performance, and reduces computation complexity by 45%.  In Section 6 of [2] empirically demonstrated that HOA can effectively approximate the optimal solution, even though it lacks formal theoretical derivation.
>
> In **A.3.4** [revised appendix, L. 1042-1068], we provide further validation. As shown in **Fig. 11** [revised appendix, L. 1045-1062], we observe that although the coherence score between tasks is not strictly symmetric ( $C_{u \to v} \neq  C_{v \to u}$), it tends to exhibit strong symmetry in practice, allowing us to approximate values while maintaining accuracy.
>
> Additionally, we mitigate the impact of task order by:
>
> - Treating coherence scores as relative measures based on gradient updates, which reduces dependency on the absolute order of task execution.
> - Averaging the coherence scores over the entire training process, which smooths out the potential variability introduced by the order of tasks.
>
>
> [2] Trevor Standley, Amir Zamir, Dawn Chen, Leonidas Guibas, Jitendra Malik, and Silvio Savarese. Which tasks should be learned together in multi-task learning? In International conference on machine learning, pp. 9120–9132. PMLR, 2020.
>
>
> ---
>
> **Q8: Is it better or worse when the same task is not allowed to appear in different clusters?**
>
> **A8:** Allowing the same task to appear in multiple clusters tends to boost performance by enabling more collaboration between tasks. In **A.3.3** [revised appendix, L.1022-1041], we conducted experiments under two setups: In Setting 1, tasks can't appear in different clusters, and in Setting 2 they can. As shown in **Table 3** [revised appendix, L.1009-1016] and **Table 4** [revised appendix, L.1026-1035], the Setting 2 achieves better performance, highlighting the benefits of task interdependence. While there’s no extra inference complexity (since each task is still only processed once during test), it does lead to higher GPU memory usage during training. So, it's a trade-off between performance gains and resource consumption.

---

> ### Author Response · Authors · 2024-12-01
> **Official Response to Reviewer mF5B (Part 5/7)**
>
> **Q9: Is there any constraint on the subsets used for set cover?**
>
> **A9:** In **A.3.5** [revised appendix, L.1249-1274], we provide implementation details for task grouping. We adopt a branch-and-bound method, similar to prior works [2], to find the optimal solution (as shown in **Algorithm 2** [revised appendix, L.1188-1219]). The prototype implementation of the algorithm can be found in [6].
>
> The following constraints govern the subsets used for set cover:
>
> 1. **Cost Budget:** Each  subset has an associated cost, represented by the latent space bitrate consumption, which is approximated by the number of groups. The total cost of the selected models cannot exceed the given budget.
>
> 2. **Task Coverage:** The selected subset must cover all tasks, meaning each task must be handled by at least one model.
>
> 3. **Performance Requirement:** For each task, the selected subset must have at least one model capable of handling it. This ensures the model can solve the task effectively.
>
> 4. **Model Rank:** Models with higher ranks (i.e., those capable of handling more tasks) are prioritized in the selection process.
>
> 5. **Performance Comparison:** A comparison is made between the losses of models to ensure that the loss of one model is not significantly worse than another across all tasks.
>
> The optimization algorithm in [6]  works within these constraints to identify the optimal subset of models.
>
> ---
>
> [2] Trevor Standley, Amir Zamir, Dawn Chen, Leonidas Guibas, Jitendra Malik, and Silvio Savarese. Which tasks should be learned together in multi-task learning? In International conference on machine learning, pp. 9120–9132. PMLR, 2020.
>
> [6] https://github.com/tstandley/taskgrouping/blob/master/network_selection/main.cpp
>
> ---
>
> **Q10: How is bitrate controlled in the proposed framework? How do you determine the weight for each task?**
>
> **A10:** The bitrate is controlled through the rate-distortion trade-off, as described by **Eq.4** [revised main page, L.156], where the term $\lambda \times \mathcal{D} + \mathcal{R}$ represents the balance between distortion ($\mathcal{D}$) and rate ($\mathcal{R}$). The parameter $\lambda$ controls this trade-off, with typical values ranging from [0.04, 0.072, 0.14, 1, 1.932]. Generally, the choice of $\lambda$ requires empirical tuning. We begin by setting $\lambda$ to 1.0, then observe the distortion compared to lossless encoding. Based on this, $\lambda$ is further adjusted to train low-bitrate and lossless models, followed by experimentation with intermediate bitrate models.
>
> For task-specific weights $w_i$ for each task $\tau_i$, we adjust them based on the task's uncompressed performance to ensure that the magnitude of the weighted loss across tasks is roughly consistent, which helps accelerate convergence during joint training. Therefore, for each task, the weight at a given bitrate is determined by $\lambda \cdot w_i$.
>
>
> ---
>
> **Q11: Is the causal discovery step required? Could the minimum description length principle be used for causal discovery and, thus, unifying step 2 and 3 as finding a DAG structure that minimizes the bit-rate?**
>
> **A11:** Thank you for your insightful comment. The causal discovery step plays a crucial role in our framework for modeling inter-task dependencies, but it is not the only possible approach. While the minimum description length (MDL) principle could be used to unify steps 2 and 3, the causal discovery step offers unique advantages by capturing specific task dependencies that are essential for optimizing compression.
> As shown in **Fig. 15** [revised appendix, L.1373-1381], we enumerate three types of task relationships:
>
> 1. **Tail-to-Tail dependency (Uniqueness)**: Tasks A and B are conditionally independent given C, but both depend on C.
> 2. **Head-to-Tail dependency (Redundancy)**: Task A influences Task B, which in turn influences Task C.
> 3. **Head-to-Head dependency（Collaborative）**: Tasks A and B each directly influence Task C.
>
> These causal relationships provide a richer understanding of the semantic structure across tasks, which is not fully captured by MDL’s focus on bitrate. Therefore, the causal discovery step complements, rather than replaces, the MDL principle by providing additional task group-aware context that enhances compression performance.

---

> ### Author Response · Authors · 2024-12-01
> **Official Response to Reviewer mF5B (Part 6/7)**
>
> **Q12: Could you provide a complexity analysis showing how computational requirements change as the number of tasks increases?**
>
> **A12:** In  **A.4** [revised appendix, L.1115-1249], we provided complexity analysis of the Lookahead Module and Taskonomy-Aware Compression Module. We also reproduce the following:
>
> 1. **Lookahead Module Complexity**
>
>    The Lookahead module consists of two main components: task grouping and DAG construction.
>
>    1.1 **Task Grouping Complexity:**
>
>    Task grouping involves several steps:
>
>    - **Computing Pairwise Coherence:** First, we compute the coherence scores for each pair of tasks. This requires evaluating $N^2/2$ task pairs, which has a time complexity of $O(N^2)$. The quadratic nature of this computation means that as the number of tasks $N$ increases, the cost grows rapidly.
>
>    - **Higher-Order Task Groupings Coherence:** After calculating the pairwise coherence, we compute the coherence of higher-order groups, which has exponential complexity represented as $O(N^N)$, where $N$ is the number of tasks. We estimate the coherence of higher-order group sets based on pairwise coherence scores. The computational overhead associated with this step is minimal and can be considered nearly negligible.
>
>    - **Optimization for Multi-Task Network Selection:** Finally, we select the optimal subset of $K$ multi-task networks from the available task groups. This step is computationally challenging, as it involves solving an NP-hard optimization problem under the given budget constraints. The complexity of this optimization problem is:
>      $$
>      O(\vert \mathcal{T} \vert \cdot \vert \boldsymbol{C}_0 \vert^{\frac{K}{Q}})
>      $$
> 		where $\vert \mathcal{T} \vert$ is the number of tasks, $\vert \boldsymbol{C}_0 \vert$ is the number of candidate networks, $K$ is the budget, and $Q$  is the minimum cost of a network in $C_0$. To solve this NP-hard problem efficiently, we employ a branch-and-bound approach. The prototype implementation of the algorithm can be found in [12].
>
>         [12] https://github.com/tstandley/taskgrouping/blob/master/network_selection/main.cpp
>
>    1.2 **DAG Construction:** After task grouping, the Lookahead module constructs a DAG based on conditional entropy. The complexity of calculating conditional and independent entropies in this stage is $O(K^2 \cdot n \cdot m)$, where $K$ is the number of task groups, $n$ is the complexity of calculating conditional entropy, $m$ is the complexity of calculating independent entropy for each group.
>
>    This part of the Lookahead module becomes computationally expensive as the number of groups grows.
>
> 	1.3. **Overall Complexity of Lookahead Module:** Although complexity increases with the number of tasks ($N$) and task groups ($K$), inspired by traditional encoding lookahead modules, using low-resolution inputs and a lightweight feature extraction backbone during pre-analysis mitigates this.
>
> 2. **Taskonomy-Aware Compression Module Complexity**
>
> 	| | **Task Encoder $l_a$**   | **Task Decoder $l_s$**     |
> 	|-------------------------------------------|----------|------------|
> 	| **KParams**                              |  16467.2  |  525.1    |
> 	| **MMACs**                                 |  25708.0  |  4968.1   |
>
> 	*Table 1: Parameters and Forward MACs of Xception-based  Encoder/Decoder of Task on 512 × 512 images.*
>
>
>
> 	| **Context Module**           | **$\boldsymbol{\theta}_{hd}$**    | **$\boldsymbol{\theta}_{cm}$**    | **$\boldsymbol{\theta}_{cm+}$**   | **$\boldsymbol{\theta}_{gc}$**    |
> 	|------------------------------|------------|------------|------------|------------|
> 	| **KParams**                  |  5810.1   |  755.2    |  1487.7   |  2264.4   |
> 	| **MMACs**                     |  8925.1   |  1148.2  |  2264.4   |  7100.5   |
>
> 	*Table 2: Parameters and Forward MACs of entropy context modules on 512 × 512 images.*
>
> 	In Taskonomy-Aware Compression Module, the process is divided into two stages: task analysis and feature compression. The feature extraction and compression  grows linearly with the number of task groups $K$. The feature decoder phase grows linearly with the number of tasks ($N$). The total complexity is:$$\text{Total Complexity of Compression} = N \cdot l_s + K \cdot (l_a+\theta_{hd}+\theta_{cm+}+\theta_{gc})$$Substituting in the specific parameters:$$\text{Total Complexity of Compression} = N \cdot 4968.1 + K \cdot 40088.0$$
>
> 3.  **Summary of Complexity**
>
>     | Module                         | Complexity                             | Dominant Term |
>     |---------------------------------|----------------------------------------|---------------|
>     | **Lookahead Module**            | $O(N^2) + O(K^2 \cdot n \cdot m)$    | $O(N^2)$    |
>     | **Taskonomy-Aware Compression  Mode**                    | $O(N \cdot l_s + K \cdot (l_a+\theta_{hd}+\theta_{cm+}+\theta_{gc}))$ | $O(N)$ |

---

> ### Author Response · Authors · 2024-12-01
> **Official Response to Reviewer mF5B (Part 7/7)**
>
> **Q13: Fig. 3, could you add a legend for what the color shade represents? Could you define the "Anchor" baseline in the Fig. caption or main text?**
>
> **A13:**  Fig. 3 in the previous manuscript corresponds to Fig. 5 [revised main paper, L.395-417] in the revised version.
>
> - The integral area of the shaded region visualizes the BD-Performance gains.
> - "Anchor" refers to the optimal performance of a supervised task obtained using uncompressed images as input.
>
> ---
>
> **Q14: How does the method perform compare to more familiar baselines such as VQGAN?**
>
> **A14:** Thank you for your thoughtful question. In **A.3.5** [revised appendix, L.1068-1114] and **Fig.12** [revised appendix, L.1080-1102], we provide an ablation study comparing our method to VQGAN, which highlights the fundamental differences in both optimization goals and performance.
>
> **1. Distinction in  Optimization Goals:** VQGAN-based compression methods [9][10] focus on optimizing perceptual compression. In contrast, our method diverges significantly by targeting efficient multi-task semantic compression. We specifically optimize for the shared use of compact semantic representations across multiple tasks, which reduces redundancy and enhances the compression efficiency.
>
> **2. Distinction in  Performance:**
> As illustrated in **Fig.12** [revised appendix, L.1080-1102], VQGAN introduces reconstruction errors due to its generative approach. These errors result in discrepancies from the ground truth, limiting its performance in tasks requiring precise feature preservation. For example, tasks like **Keypoint 2D detection**  require capturing fine-scale, sparse local features. These tasks are challenging for VQGAN, as its generative nature is less suited to applications that require high precision.
> In contrast, our method not only achieves superior compression efficiency at low bitrates but also maintains near-optimal performance at higher bitrates, closely aligning with the supervision anchor.
>
> In conclusion, the key advantage of our approach lies in its ability to compress semantic information across tasks required for high precision. VQGAN is based on a generative framework and focuses on perceptual quality results in inferior fidelity performance.
>
> ---
>
> [9] Patrick Esser, Robin Rombach, and Bjorn Ommer. Taming transformers for high-resolution image
> synthesis. In Proceedings of the IEEE/CVF conference on computer vision and pattern recognition,
> pp. 12873–12883, 2021.
>
> [10] Qi Mao, Tinghan Yang, Yinuo Zhang, Zijian Wang, Meng Wang, Shiqi Wang, Libiao Jin, and Siwei
> Ma. Extreme image compression using fine-tuned vqgans. In 2024 Data Compression Conference
> (DCC), pp. 203–212. IEEE, 2024.
>
> ---
>
> **Q15: Github link is not provided?**
>
> **A15:** Thank you for your interest in accessing the code. We will include it in the final manuscript once it is fully prepared.
>
>
> ---
>
> **Q16: Typo error.**
>
> **A16:** Thank you for pointing that out. We will correct the typo in the manuscript.

---

> > ### Comment · Reviewer_mF5B · 2024-12-03
> >
> > Thanks for the authors' detailed reply. The paper is updated with significant changes. I think I need a bit more time to parse through the revision before reaching a conclusion.

---

> > > ### Author Response · Authors · 2024-12-03
> > >
> > > Thank you for taking the time to review our paper. We've carefully considered your initial feedback and worked on making our methodology clearer. We've also added more experiments and analyzed the results, which led to some interesting and insightful findings. We really appreciate your help in improving our work.

---

### Official Review · Reviewer_63Jo · 2024-11-03

**Soundness:** 4
**Presentation:** 3
**Contribution:** 4
**Rating:** 8
**Confidence:** 3

**Summary:**

In this paper, the authors focus on learning methods for image compression. To this end, they propose a method that optimizes image compression for a number of different downstream visual tasks. They consider the potential redundancy of encodings for similar groups of visual tasks, and they utilize directed acyclic graphics to learn causal relationships between tasks. This approach leads to a better multi-task representation. They evaluate their approach on a number of different visual tasks and demonstrate convincing quantitative results.

**Strengths:**

1. The problem has multiple applications and is very sensible. Image compression has direct implications, and it is sensible to think image compression in terms of its final use (object detection, classification, etc) as opposed to only pixel reconstruction.

2. The proposed approach has a strong mathematical foundation and makes intuitive sense (conditional entropy should help resolve redundancy between certain tasks).

3. Quantitative results are thorough and show promise.

**Weaknesses:**

1. I believe the description of the DAG learning should have more detail. Specifically, some of the details of the DAG based algorithm (described in the appendix as Algorithms 1 and 2) should be incorporated into the main paper.

2. There are some typos and grammatical errors.

**Questions:**

1. Would it be possible to incorporate more details of the graph learning into the main paper?

---

> ### Author Response · Authors · 2024-12-01
> **Official Response to Reviewer 63Jo**
>
> Dear Reviewer 63Jo,
>
> Thank you for the positive feedback! We explored how semantic representations at different granularities exhibit redundancy, uniqueness, and collaborative causal relationships. We believe that dependencies and redundancies among multiple tasks should be considered to optimize compression for downstream tasks, rather than focusing solely on pixel-level fidelity.
>
> In the revised version, we added **Figure 1(c)**[revised main paper, L.37-42], which visually demonstrates the redundancy and uniqueness of latent representations across different tasks.
> **Task Grouping** can effectively leverage semantic collaboration, maximizing information sharing between related tasks.
> Additionally, **Causal Discovery via Conditional Entropy** models causal relationships between tasks, allowing child task representations to gain contextual information from the parent task layers, thereby reducing redundancy and uncertainty.
>
> ---
> **Q1: Is it possible to incorporate more details of the graph learning (DAG learning) into the main paper rather than just in the appendix?**
>
> **A1:** We agree with your suggestion that incorporating additional details of the DAG-based algorithm into the main paper would help clarify our approach. In the revised version, we've revised as follows:
> - We added a section titled **Constructing DAG via Conditional Entropy** [revised main paper, L.282-311], which includes **Algorithms 1** [previous appendix, L.925-943] on DAG construction into the main text [revised main paper, L.290-309].
>
> - For **Algorithm 2** [previous appendix, L.946-965], we have added a section titled **Scalable Compression Using the Causal DAG** [revised main paper, L.312-357] that explains in detail how to execute compression by traversing the graph in topological order after constructing DAG. The implementation details show how the parent representation $\boldsymbol{y}_p$ improves compression efficiency by providing the child representation $\boldsymbol{y}_c$ with an extra cross-task context.
>
> ---
> **Q2: Typos and grammatical errors.**
>
> **A2:** Thanks for pointing out the issues. We have corrected the typos and grammatical errors in the revised manuscript, which should have enhanced its overall presentation and clarity.

---

### Author Response · Authors · 2024-12-03
**Review and Rebuttal Summary**

We would like to thank all the reviewers for their time and valuable comments.

We are happy that all three reviewers recognize our method as theoretically grounded and impactful, highlighting its applicability to compression for automated machine vision analytics and may require occasional human viewing, such as the Internet of Things, smart cities, and multimodal large models. Reviewers 63Jo and mF5B commend the strong performance gains, reviewers 63Jo and 5H7f highlight the innovation and broader applicability of our hierarchical representations and causal modeling. **By leveraging disentangled representations, task grouping, and causal links, our method efficiently facilitates collaboration, uniqueness, and conflict resolution when compressing for complicated tasks.**

We summarize the main concerns and corresponding actions taken during the rebuttal:

1. **Improve the clarity of the framework, especially Taskonomy-Aware Compression:**
   - We added the background and formal definition of the compression problem (**Related Work**), an overview of how to extend end-to-end compression to Taskonomy-Aware Compression (**Fig. 2**), and more detailed implementation details (**Sec 3.3**).

2. **Improve the clarity of the DAG module, especially Algorithms 1 and 2 in the previous appendix:**
   - We added a subsection **Constructing DAG via Conditional Entropy** [revised main paper, L.282-311] and a subsection **Scalable Compression Using the Causal DAG** [revised main paper, L.312-357] to enhance clarity.

3. **Add additional ablation experiments to validate the effectiveness of the proposed method:**
   - We added end-to-end compression with task auxiliary loss to demonstrate **the existence of task collaboration, conflict, and uniqueness, as well as the necessity of feature disentangle and task grouping.**(**A.3.1**)
   - We added an experiment "No task grouping, treat each task independently" to show that **most tasks benefit from task grouping in both accuracy and compactness. An additional benefit is that grouped tasks only need to perform encoding once.** (**A.3.2**)
   - **Robustness of the task grouping module:**
     - Impact of not allowing the same tasks in different clusters. (**A.3.3**)
     - Impact of task order. (**A.3.4**)
   - Comparison to VQGAN to clarify the distinct optimization goals and significant performance divergence. (**A.3.5**)

4. **Complexity Analysis:**
   - We added a complexity analysis of Parameter Sets, Task Grouping, DAG Construction, and Taskonomy-Aware Compression. (**A.4**)

5. **Implementation details of task grouping:**
   - We provided a pseudo-algorithm and the prototype implementation. (**A.5**)

---

### Meta-Review · Area_Chair_Tu6d · 2024-12-24

**Metareview:**

Reviewers consider the studied problem highly significant and sensible, with multiple immediate applications. The proposed approach is considered to be both well-motivated mathematically and intuitively sensible. Reviewers are also in agreement that results demonstrate strong performance vis-a-vis end-to-end compression methods using the same bitrate. Performance on downstream tasks consistently outperforms trained compression methods (e.g. MLIC++, ELIC).

While several criticisms have been raised during the initial reviewer feedback (with clarity being a major concern), the authors provided a detailed and thorough rebuttal addressing several concerns. Unfortunately, several reviewers failed to respond to this detailed rebuttal, forgoing what I believe ought to have been a likely score increase. Having re-read this submission carefully, I believe that this submission (provided the thorough rebuttal is included in a camera-ready version) is currently above the acceptance threshold, and the raw scores themselves are somewhat undeservingly low, given the non-responsiveness of the reviewers mentioned above. As such, I recommend acceptance.

**Additional Comments On Reviewer Discussion:**

Reviewer Discussion for this submission has unfortunately been disappointing despite multiple reminders. Reviewers mF5B and 5H7f failed to respond to an exemplary and detailed rebuttal which clearly addressed several of the criticisms, providing both written responses, formal explanations and experimental results (as reviewer mF5B acknowledges). While it is impossible to say what both reviewers would have concluded, I think it is likely to assume that a score increase may have been fair in both circumstances, especially given the low initial score by 5H7f.

---

### Decision · Program_Chairs · 2025-01-22

Accept (Poster)